# Selenium supplementation inhibits IGF-1 signaling and confers methionine restriction-like healthspan benefits to mice

Jason D Plummer[1], Spike DL Postnikoff[2], Jessica K Tyler[2], Jay E Johnson[1]*

[1]Department of Biology, Orentreich Foundation for the Advancement of Science, Cold Spring, United States; [2]Department of Pathology and Laboratory Medicine, Weill Cornell Medicine, New York, United States

**Abstract** Methionine restriction (MR) dramatically extends the healthspan of several organisms. Methionine-restricted rodents have less age-related pathology and increased longevity as compared with controls, and recent studies suggest that humans might benefit similarly. Mechanistically, it is likely that the decreased IGF-1 signaling that results from MR underlies the benefits of this regimen. Thus, we hypothesized that interventions that decrease IGF-1 signaling would also produce MR-like healthspan benefits. Selenium supplementation inhibits IGF-1 signaling in rats and has been studied for its putative healthspan benefits. Indeed, we show that feeding mice a diet supplemented with sodium selenite results in an MR-like phenotype, marked by protection against diet-induced obesity, as well as altered plasma levels of IGF-1, FGF-21, adiponectin, and leptin. Selenomethionine supplementation results in a similar, albeit less robust response, and also extends budding yeast lifespan. Our results indicate that selenium supplementation is sufficient to produce MR-like healthspan benefits for yeast and mammals.

*For correspondence:
jjohnson@orentreich.org

## Introduction

It has been well established that methionine restriction (MR) can improve mammalian healthspan. For example, rats fed a methionine-restricted diet are substantially longer-lived than their control-fed counterparts and show a marked amelioration of age-related pathologies; methionine-restricted mice receive similar benefits (*Miller et al., 2005*; *Orentreich et al., 1993*; *Richie et al., 1994*). Among the varied benefits of MR to rodents, there is an improvement in metabolic health, marked by reduced white adipose tissue accumulation, amelioration of liver steatosis (aka, 'fatty liver disease'), and improved glycemic control (*Malloy et al., 2006*; *Malloy et al., 2013*; *Miller et al., 2005*). In fact, these metabolic benefits are so robust that MR provides complete protection against diet-induced obesity, which results from feeding animals a high-fat diet meant to approximate the human Western diet (*Ables et al., 2012*). In addition, as part of the multi-faceted response to MR, restricted animals also demonstrate altered plasma levels of the nutrient- and stress-sensing hormones IGF-1, FGF-21, adiponectin, and leptin.

Recent studies have suggested that the response to MR is conserved throughout phylogeny and that methionine-restricted humans are likely to receive similar healthspan benefits to rodents (*Dong et al., 2018*; *Gao et al., 2019*; *Johnson and Johnson, 2014*). As the vegan diet is naturally low in proteins and free amino acids, a methionine-restricted diet is technically feasible for humans (*McCarty et al., 2009*). However, such a diet might not be practical for all individuals, and as a result, widespread adherence is likely to be problematic. In addition, consumption of a diet low in amino acids other than methionine and cysteine, both of which must be restricted for efficient MR,

might result in undesirable side effects. Thus, an obvious goal of the healthy aging field is the identification and/or development of interventions that produce the benefits associated with MR, but in the context of a normal, methionine-replete diet.

An important clue for the identification and development of an MR-like intervention that is effective in a methionine-replete context is the observation that a decrease in the circulating levels of the energy-regulating hormone IGF-1 (*Ables et al., 2012*; *Malloy et al., 2006*) is likely responsible for many, if not all, of the health benefits of MR. Specifically, in an exhaustive study comparing the effects of MR and impairment of growth hormone (GH)/IGF-1 signaling on healthspan, it was found that feeding a methionine-restricted diet to long-lived dwarf mice with low IGF-1 levels failed to further extend the lifespan of these animals (*Brown-Borg et al., 2014*). Given that MR reduces circulating IGF-1 levels, the lack of either additive or synergistic effects of MR and GH/IGF-1 impairment on overall longevity suggests that MR extends healthspan primarily (and possibly even entirely) by lowering IGF-1 levels. Indeed, unpublished studies from the Orentreich Foundation found that GH injections abrogate the MR phenotype of methionine-restricted rats (not shown). In addition, Brown-Borg et al. recently demonstrated that short-lived GH-overexpressing transgenic mice show an impaired response to MR (*Brown-Borg et al., 2018*). Accordingly, we hypothesized that any intervention that reduces IGF-1 levels (or otherwise impairs IGF-1 signaling) will produce MR-like healthspan benefits. Indeed, in addition to MR, many other healthspan-extending dietary interventions are known to feature impaired IGF-1 signaling, including calorie restriction (CR), intermittent fasting, ketogenic diet, and protein restriction (*Ables et al., 2012*; *Boden et al., 2005*; *Breese et al., 1991*; *Dunn et al., 1997*; *Fontana et al., 2008*; *Fraser et al., 2000*; *Malloy et al., 2006*; *Miller et al., 2005*; *Rahmani et al., 2019*; *Scarth, 2006*). While future studies will determine to what extent impaired GH/IGF-1 signaling mediates the improved healthspan associated with these interventions, the fact that they all feature reduced IGF-1 levels provides strong evidence that decreased IGF-1 signaling mediates the health benefits of these regimens, and further, may actually be sufficient to extend healthspan.

Prompted by the observation that selenium supplementation limits the growth of young rats, as well as the fact that selenium can accumulate in the pituitary, Thorlacius-Ussing et al. explored whether GH/IGF-1 signaling might be impaired by this intervention (*Thorlacius-Ussing et al., 1987*). The authors found that plasma levels of GH were reduced by 77% in rats exposed to sodium selenite as compared with control animals. Additionally, as would be expected given the fact that GH signaling controls the release of IGF-1 into the bloodstream, circulating IGF-1 levels were also reduced by 83% in selenium-supplemented animals. Not only does this finding explain the small body size of selenium-supplemented rats, but the fact this intervention dramatically reduces IGF-1 levels raises the possibility that, by doing so, it might also confer MR-like healthspan benefits.

To test the hypothesis that selenium supplementation might confer MR-like benefits to mice by reducing circulating IGF-1 levels, we fed an otherwise normal high-fat diet containing sodium selenite to mice and assessed whether, like MR, this intervention protects against diet-induced obesity. To confirm that sodium selenite supplementation acted as expected, we measured circulating IGF-1 levels. We also assessed multiple other physiological parameters known to be altered by MR, as well as the ability of other selenium sources to support any benefits associated with sodium selenite supplementation. Here, we show that selenium supplementation confers a variety of healthspan benefits typically associated with MR to mice, including dramatically lower white adipose tissue accumulation, improved glycemic control, and altered plasma levels of IGF-1, FGF-21, adiponectin, and leptin. Moreover, we present data from yeast studies that suggest a potential mechanism underlying these benefits. Taken together, our results indicate that selenium supplementation produces the healthspan benefits associated with MR, but in a normal, methionine-replete context.

## Results

### Like MR, sodium selenite supplementation protects male and female mice against diet-induced obesity

In order to determine whether, like MR and other healthspan-extending interventions, selenium supplementation might protect mice against diet-induced obesity, we fed three isocaloric synthetic diets to wild-type male C57BL/6J mice for a period of at least 16 weeks and assessed multiple parameters

representative of their general body condition and metabolic health. Three so-called 'high-fat' diets were formulated so as to provide 57% of total calories from fat (as compared with 10% for a typical synthetic diet) and were as follows: (1) a normal control diet containing adequate methionine (0.86%; CF), (2) a methionine-restricted diet (0.12%; MR), and (3) an otherwise normal, methionine-replete diet, but containing 0.0073% sodium selenite (CF-SS). Amounts of selenium-containing compounds used for this and subsequent experiments were determined from pilot studies aimed at identifying minimal doses that still achieve high efficacy (not shown). To assess the extent of protection against diet-induced obesity, body mass and food consumption were recorded once per week until the end of the experiment (*Figure 1A–C*), at which time, animals were sacrificed and their overall body condition assessed (*Figure 1D–F*). After these measurements, inguinal and perigonadal fat pads were surgically resected and weighed (*Figure 1G,H*). Intact livers, which are susceptible to fat

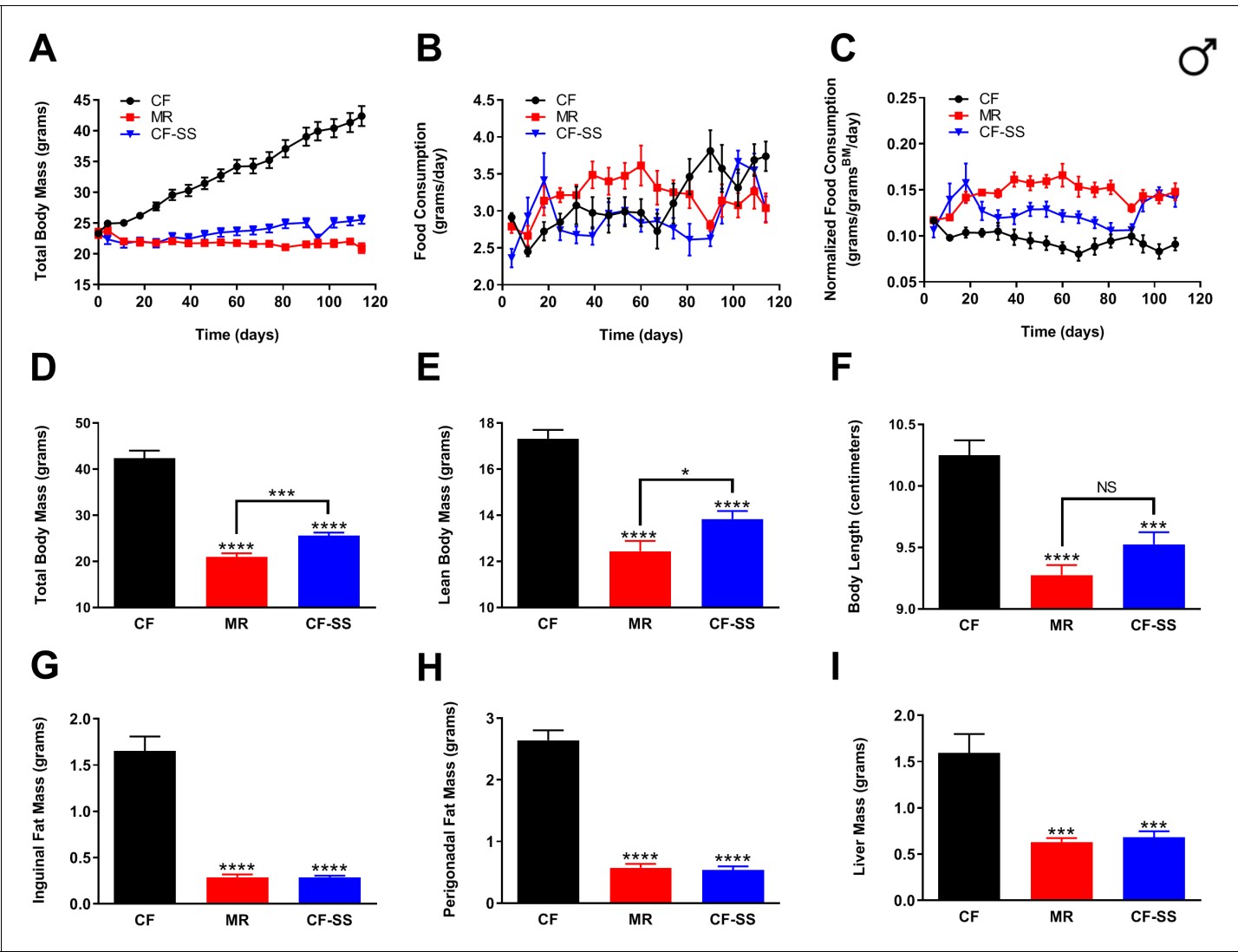

**Figure 1.** Sodium selenite supplementation protects male mice against diet-induced obesity. Comparisons over time of average values for (A) total body mass, (B) food consumption, and (C) food consumption normalized to total body mass for control-fed (CF; black circles), methionine-restricted (MR; red squares), and sodium selenite-supplemented (CF-SS; blue triangles) male mice. Average values at conclusion of the experiment (16.3 weeks) are also shown for (D) total body mass, (E) lean body mass, (F) body length, (G) mass of inguinal fat pads, (H) mass of perigonadal fat pads, and (I) liver mass for all feeding conditions. For all panels, bars denote standard error of the mean (SEM). For panels D–I, statistically significant differences (as compared with the corresponding CF values) are indicated (***$p < 0.001$; ****$p < 0.0001$). N = 8 for all groups.

The online version of this article includes the following figure supplement(s) for figure 1:

**Figure supplement 1.** Fat depot and liver sizes normalized to lean body mass for sodium selenite-supplemented male mice.

accumulation in animals fed a high-fat diet (*Malloy et al., 2013*), were also obtained and weighed (*Figure 1I*). Using this approach, we found that sodium selenite supplementation completely protected male mice against the dramatic weight gain observed for control-fed animals (*Figure 1A,D*). In fact, the protection against diet-induced obesity conferred by sodium selenite supplementation was essentially identical to that observed for methionine-restricted male mice (*Figure 1A,D,G,H*). In particular, both interventions resulted in a robust reduction in the accumulation of inguinal (MR, 82%; CF-SS, 83%) as well as perigonadal (MR, 78%; CF-SS, 80%) adipose tissue (*Figure 1G,H*). At the end of the experiment (16 weeks), the average total body mass of sodium selenite-supplemented animals was slightly higher than that of methionine-restricted animals (25.6 g vs 21.0 g; *Figure 1A,D*), although this difference was partly due to the greater lean body mass observed for sodium selenite-supplemented males as compared with methionine-restricted counterparts (13.8 g vs 12.4 g; *Figure 1E*). Put another way, the growth inhibitory effect known to be associated with MR (*Ables et al., 2012*; *Miller et al., 2005*; *Orentreich et al., 1993*) was somewhat less pronounced for sodium selenite supplementation. Furthermore, this lesser growth inhibition was consistent with measurements of the overall body length of the animals. That is, sodium selenite-supplemented male mice might have been somewhat longer than their methionine-restricted counterparts (9.52 cm vs 9.28 cm; *Figure 1F*). However, the observed difference is not quite statistically significant (p=0.07). Nevertheless, methionine-restricted and sodium selenite-supplemented animals were equally resistant to the accumulation of both inguinal and perigonadal adipose tissue (*Figure 1G,H*), even when differences in lean body mass were taken into account (*Figure 1—figure supplement 1A,B*). In addition, the smaller liver weights resulting from these two interventions were also nearly identical (0.63 g vs 0.68 g; *Figure 1I*, *Figure 1—figure supplement 1C*), suggesting that sodium selenite-supplemented male mice were, like their methionine-restricted littermates, protected against liver steatosis. To confirm that the observed differences in fat accumulation and body condition in sodium selenite-supplemented animals were due directly to the presence of the compound rather than a putative calorie reduction in the event that animals found the food unpalatable, we also assessed the rate of food consumption for all three diets. With respect to absolute food consumption, sodium selenite-containing food was consumed equivalently to the control diet (*Figure 1B*), whereas methionine-restricted food was consumed at a greater rate, consistent with previous findings. When normalized to body size, the consumption of sodium selenite-containing food was greater than that of the control diet, albeit somewhat less than that of methionine-restricted food (*Figure 1C*). Thus, male mice found sodium selenite-containing food to be just as palatable as the control diet and were not calorie-restricted. This confirms that the total protection against diet-induced obesity enjoyed by male mice was directly due to sodium selenite supplementation.

To test whether female mice might benefit similarly from this intervention, a group of females was subjected to the same feeding regimens as described above for males. Measurements of body mass and food consumption were performed longitudinally (*Figure 2A–C*), determination of multiple metrics of body condition was performed at termination (*Figure 2D–F*), and assessment of adiposity was performed by weighing surgically resected fat pads and liver (*Figure 2G–I*). However, while the male mice used for these experiments were relatively young adults (2 months), we made use of older adult females (9 months) as our empirical observations (not shown) have demonstrated that young female mice remain relatively lean and metabolically unaffected by a high-fat synthetic diet. Using 9-month-old female mice for these studies, we found that females of this age dramatically increased in weight (*Figure 2A,D*) and readily accumulated adipose tissue (*Figure 2G,H*) when fed a high-fat diet. In contrast, MR ameliorated the diet-induced obesity of female mice (*Figure 2A,D,G,H*). To our knowledge, this is the first time that this has been reported for wild-type females. In any case, we were surprised to observe that not only does sodium selenite supplementation protect female mice against diet-induced obesity, it did so arguably more effectively than MR. Whereas all animals were weight-matched at the beginning of the study, sodium selenite-supplemented female mice not only had dramatically less body mass than control-fed littermates, but they were also marginally lighter than methionine-restricted animals throughout the experiment (*Figure 2A*). Furthermore, sodium selenite-supplemented animals demonstrated more robust decreases in both inguinal adiposity (81% less than controls) and perigonadal adiposity (80% less than controls) than did methionine-restricted littermates (53% and 56% less than controls, inguinal and perigonadal; *Figure 2G,H*, *Figure 2—figure supplement 1A,B*). Interestingly, and in contrast to the case for males, effects of MR and sodium selenite supplementation were similar with respect to lean body mass and body length (*Figure 2E*,

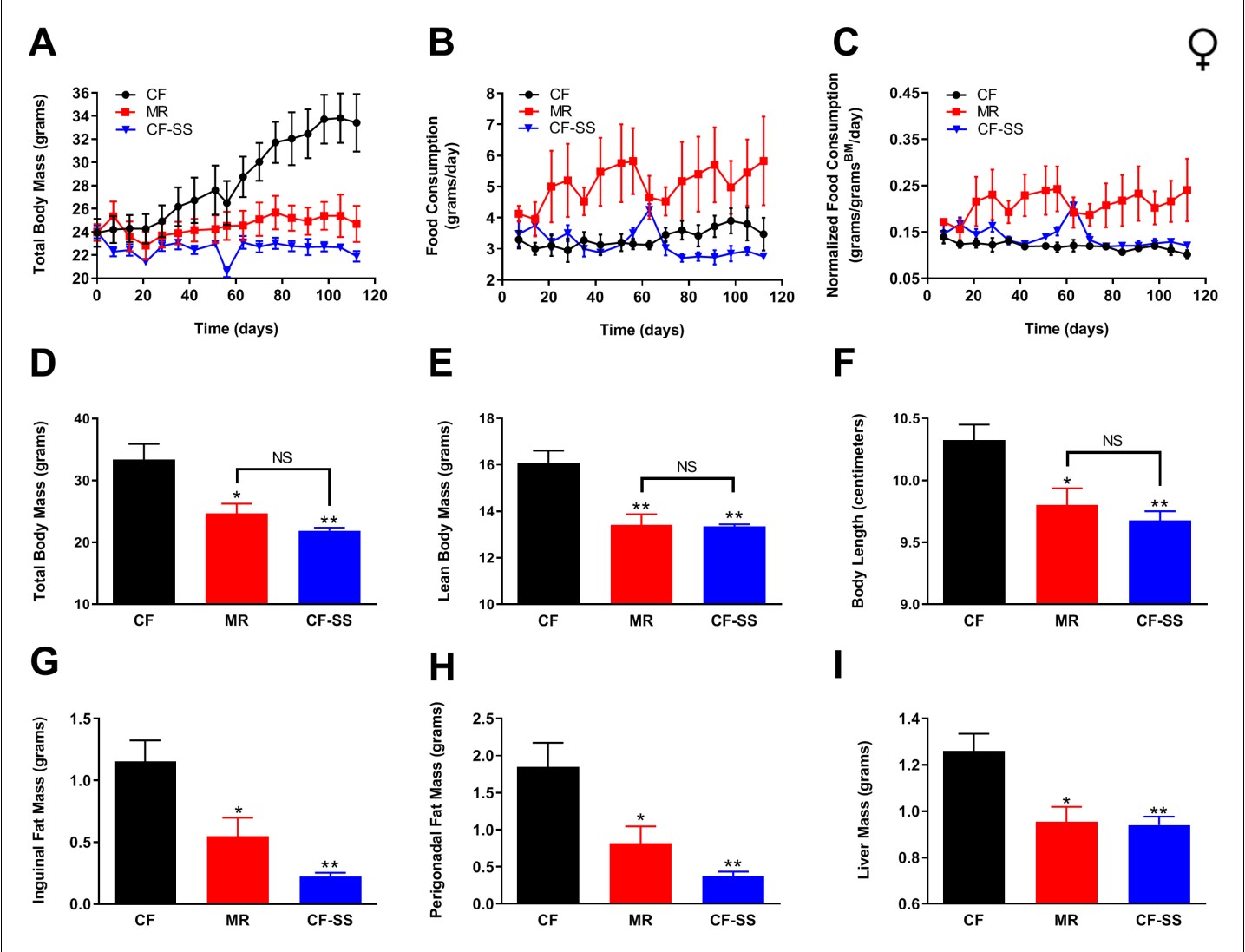

**Figure 2.** Sodium selenite supplementation protects female mice against diet-induced obesity. Comparisons over time of average values for (**A**) total body mass, (**B**) food consumption, and (**C**) food consumption normalized to total body mass for control-fed (CF; black circles), methionine-restricted (MR; red squares), and sodium selenite-supplemented (CF-SS; blue triangles) female mice. Average values at conclusion of the experiment (16 weeks) are also shown for (**D**) total body mass, (**E**) lean body mass, (**F**) body length, (**G**) mass of inguinal fat pads, (**H**) mass of perigonadal fat pads, and (**I**) liver mass for all feeding conditions. For all panels, bars denote SEM. For panels D–I, statistically significant differences (as compared with the corresponding CF values) are indicated (*p<0.05; **p<0.01). N = 4 for all groups.

The online version of this article includes the following figure supplement(s) for figure 2:

**Figure supplement 1.** Fat depot and liver sizes normalized to lean body mass for sodium selenite-supplemented female mice.

*F*). This difference might be due to the fact that the female mice were older than the males (9 months old vs 2 months old) and nearly fully grown at the start of the intervention period. Alternatively, it is possible that sodium selenite supplementation has sex-specific effects on lean body mass, whereas MR does not. In any case, the efficacy of sodium selenite supplementation in protecting female mice against adipose accumulation is clear; and this effect may extend to fatty liver, as liver mass was lower in both methionine-restricted (24%) and sodium selenite-supplemented (25%) females as compared with obese control animals (*Figure 2I*, *Figure 2—figure supplement 1C*). As a point of interest, while the livers of methionine-restricted and sodium selenite-supplemented female mice were slightly larger than those of their male counterparts (~0.95 g vs ~ 0.65 g), this difference is likely due to the fact that these animals were older than their male counterparts (9 months old vs 2 months old), and thus, nearly fully grown. Finally, and similar to the case for males, the smaller body

size and lower adiposity of sodium selenite-supplemented female mice were not due to reduced calorie intake, as their food consumption was at least equivalent to control-fed animals, whether normalized for body weight or not (*Figure 2B,C*). In total, these experiments clearly demonstrate that sodium selenite supplementation is sufficient to completely protect both male and female mice against diet-induced obesity.

## Sodium selenite supplementation results in a variety of physiological, hormonal, and metabolic changes typically associated with MR

In order to (1) confirm that sodium selenite supplementation decreases plasma IGF-1 levels in C57BL/6J mice, and (2) test for additional similarities between this intervention and MR, we assessed multiple circulating analytes from male and female mice fed the control, methionine-restricted, and sodium selenite-supplemented diets. For this purpose, we obtained blood samples from animals at baseline (i.e., before initiation of the experimental diets), as well as after 4 weeks, 8 weeks, and 16 weeks on diet (with the last time-point representing the conclusion of the experiment). Plasma samples were analyzed by ELISAs to determine the concentrations of IGF-1, FGF-21, leptin, and adiponectin. These particular analytes were selected because they are not only affected by MR (and thus, constitute a subset of the 'MR phenotype') (*Ables et al., 2012*; *Malloy et al., 2006*), but have also been shown to be involved in the regulation of metabolism and/or healthspan. For example, circulating levels of the energy-regulating hormones adiponectin and leptin are increased and decreased, respectively, by MR, presumably due to the reduction of adiposity associated with this intervention (*Ables et al., 2012*; *Elshorbagy et al., 2011*; *Malloy et al., 2006*). In addition, the hepatokine FGF-21 is increased by MR and not only has roles in glucose metabolism (*Kharitonenkov et al., 2005*), but might also participate in the regulation of longevity via interaction with the IGF-1 receptor and its co-receptor β-Klotho (*Kurosu et al., 2005*; *Liu et al., 2007*). Also, as mentioned previously, plasma levels of the energy-regulating, anabolic hepatokine IGF-1 are reduced by MR in rodents (*Ables et al., 2012*; *Malloy et al., 2006*), and such reduction is apparently sufficient to confer extended healthspan (*Brown-Borg et al., 2014*). That said, we expected that IGF-1 levels should be reduced by sodium selenite supplementation based on results from studies in rats (*Thorlacius-Ussing et al., 1987*). Indeed, we found plasma IGF-1 levels to be dramatically reduced in male mice by both MR and sodium selenite supplementation (*Figure 3A*). At the conclusion of the experiment, IGF-1 levels were reduced by 53% (MR) and 37% (CF-SS), respectively, as compared with control values; similar reductions were seen at all time-points following initiation of feeding the experimental diets. Plasma levels of leptin followed a similar trend, with concentrations reduced equivalently by MR and sodium selenite supplementation at all time-points (*Figure 3B*). As an example, male mice that had been fed methionine-restricted and sodium selenite-supplemented diets for 16 weeks showed reductions in circulating leptin of 98% and 96%, respectively, as compared with controls. In contrast, MR-like changes in the plasma levels of adiponectin and FGF-21 were not apparent for sodium selenite-supplemented males. In other words, at all time-points assessed, the levels of adiponectin and FGF-21 in the plasma of sodium selenite-supplemented male mice did not show the increases typically observed for MR (*Figure 3C,D*) and were not significantly different from those of control animals. While these represent the first phenotypic differences that we observed for methionine-restricted and sodium selenite-supplemented animals, subsequent experiments revealed that FGF-21 is indeed responsive to sodium selenite, just not under all conditions. This is discussed in greater detail below. The final circulating analytes assessed for male mice were glucose and insulin, both of which were significantly reduced by MR and sodium selenite supplementation (*Figure 3E,F*). This finding suggests that the beneficial metabolic milieu conferred to male mice by selenium supplementation also includes improved glycemic control and is consistent with a previous report that selenium can regulate glucose metabolism and may be an 'insulin mimetic' (*Stapleton, 2000*).

With respect to female mice, the trend of them demonstrating a greater response to sodium selenite supplementation than to MR was maintained for many of the circulating analytes assayed. For example, IGF-1 levels were decreased by sodium selenite supplementation as compared with control animals and were lower even than those resulting from MR (*Figure 4A*). The most robust effect was observed at 8 weeks, when the concentration of IGF-1 was decreased by 65% as compared with controls. The reduction resulting from MR at the same time-point was only 33%. Similarly, leptin levels in female mice were reduced by both interventions (*Figure 4B*). At the terminal time-point, leptin levels were 87% (CF-SS) and 63% (MR) lower than control levels. Interestingly, both MR and

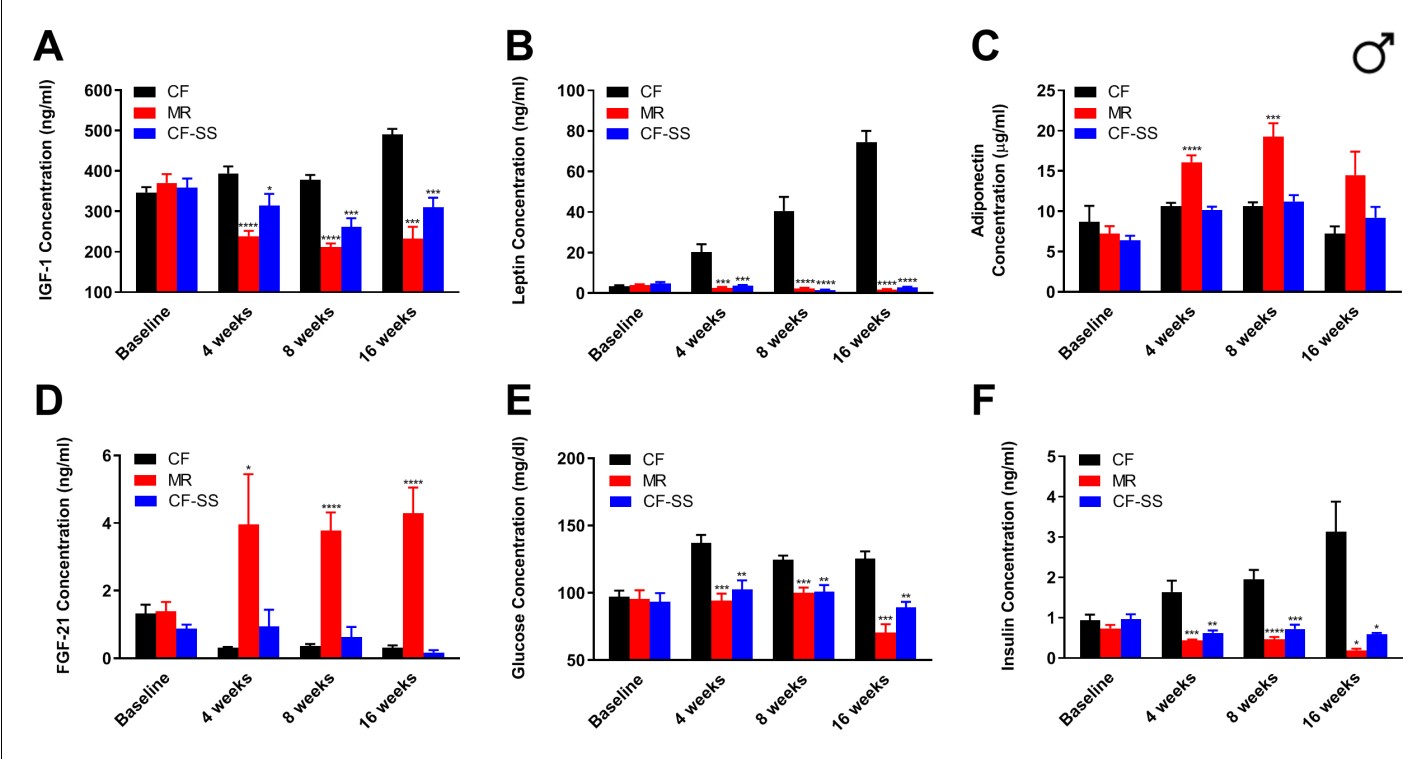

**Figure 3.** Sodium selenite supplementation decreases IGF-1 levels in male mice and results in beneficial plasma hormone and chemical changes typically associated with MR. Longitudinal comparisons of the plasma concentrations of (A) IGF-1, (B) leptin, (C) FGF-21, (D) adiponectin, (E) glucose, and (F) insulin for control-fed (CF), methionine-restricted (MR), and sodium selenite-supplemented (CF-SS) male mice. Bars denote SEM. Statistically significant differences (as compared with the corresponding CF values) are indicated (*p<0.05; **p<0.01; ***p<0.001; ****p<0.0001). N = 8 for all groups.

sodium selenite supplementation resulted in increases of adiponectin and FGF-21 levels in female mice (*Figure 4C,D*). After consuming the diets for 8 weeks, animals demonstrated 34% (CF-SS) and 30% (MR) higher levels of adiponectin than control mice, as well as 40-fold (CF-SS) and 17-fold (MR) higher levels of FGF-21. In addition, sodium selenite-supplemented females also showed improved glycemic control, as both plasma glucose and plasma insulin levels were significantly reduced by this intervention at the 8-week and 16-week time-points (*Figure 4E,F*). Thus, by all tested metrics, sodium selenite was found to recapitulate phenotypes associated with MR when fed to female mice. In consideration of why this was not found to be the case for male mice, which did not demonstrate altered plasma adiponectin and FGF-21 levels upon sodium selenite supplementation, we explored whether the older age of the females used for these studies (9 months old vs 2 months old for males) might account for this difference. Toward this end, we fed both the control and sodium selenite-containing diets to older adult male mice (8 months old) and assessed body mass and food intake, as well as the plasma levels of FGF-21 and adiponectin (*Figure 4—figure supplement 1*). While these mice demonstrated both normal food consumption and the protection against diet-induced obesity (*Figure 4—figure supplement 1A–C*) described above for 2-month-old male mice, they also showed a 30-fold higher level of circulating FGF-21 as compared with control-fed littermates, and this after only 8 weeks on diet (*Figure 4—figure supplement 1D*). Therefore, it is both notable and apparent that the regulation of FGF-21 by sodium selenite supplementation is age-dependent. In contrast, adiponectin levels in male mice remained unaffected by this intervention, even in 8-month-old animals (*Figure 4—figure supplement 1E*). Thus, the latter finding suggests that selenium supplementation-dependent alterations in the plasma levels of adiponectin are sex-specific. In any case, taken together, our results indicate that sodium selenite supplementation produces most, if not all, of several MR-associated beneficial physiological changes in both male and female mice.

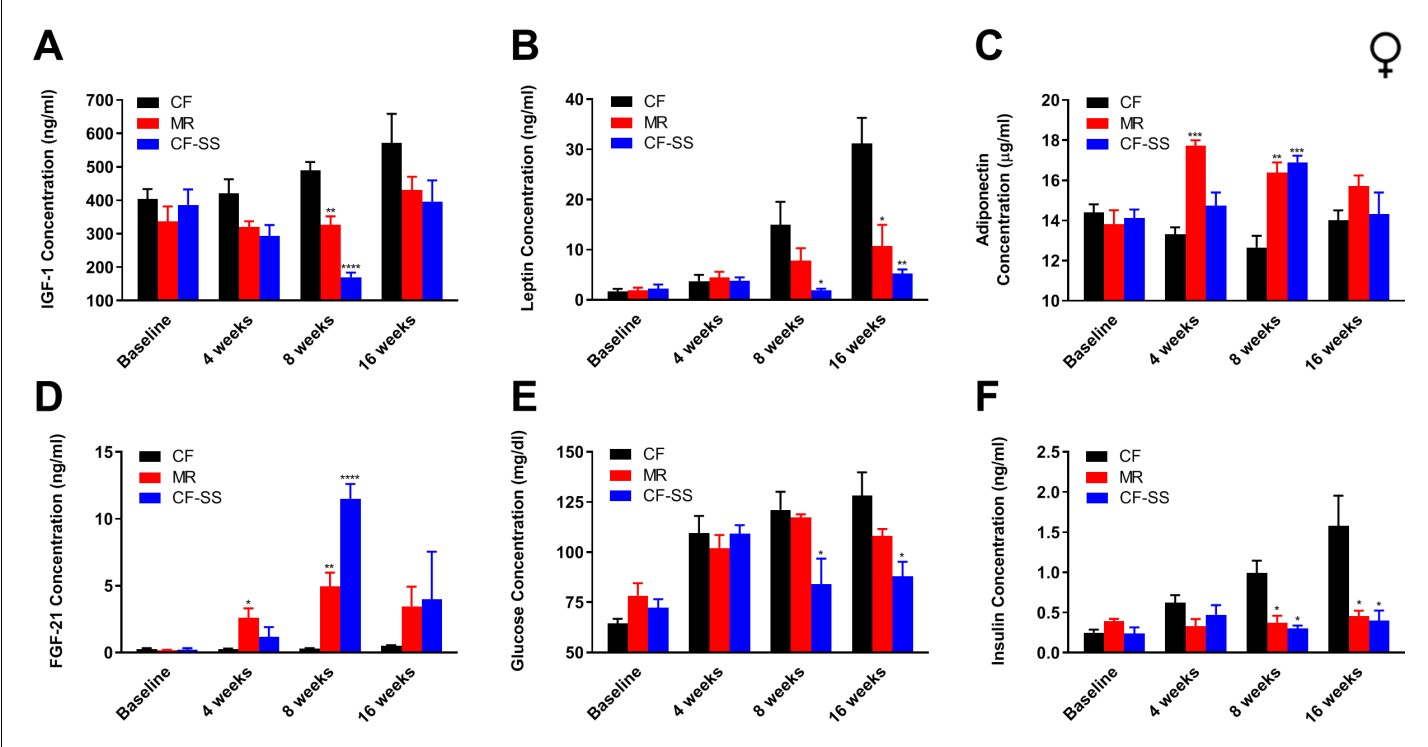

**Figure 4.** Sodium selenite supplementation decreases IGF-1 levels in female mice and results in beneficial plasma hormone and chemical changes typically associated with MR. Longitudinal comparisons of the plasma concentrations of (**A**) IGF-1, (**B**) leptin, (**C**) FGF-21, (**D**) adiponectin, (**E**) glucose, and (**F**) insulin for control-fed (CF), methionine-restricted (MR), and sodium selenite-supplemented (CF-SS) female mice. Bars denote SEM. Statistically significant differences (as compared with the corresponding CF values) are indicated (*p<0.05; **p<0.01; ***p<0.001; ****p<0.0001). N = 4 for all groups.

The online version of this article includes the following figure supplement(s) for figure 4:

**Figure supplement 1.** Sodium selenite supplementation-mediated increases in circulating FGF-21 levels are age-dependent.

## Sodium selenite supplementation reduces plasma IGF-1 levels without evidence of a reduction in GH levels

Even though, as hypothesized, we found that sodium selenite supplementation reduced circulating IGF-1 levels, it remained unknown whether or not this reduction was a consequence of reduced GH production. As mentioned above, selenium supplementation results in reduced GH release in rats, which in turns reduces the levels of circulating IGF-1 (*Thorlacius-Ussing et al., 1987*). To determine whether or not impairment of IGF-1 signaling in selenium-supplemented mice is associated with reduced GH, we measured the plasma GH levels of young male mice that had been fed the control and sodium selenite-containing diets for 16 weeks. As GH is normally released into the bloodstream from the anterior pituitary in a pulsatile fashion, we expected to detect relatively low concentrations of this hormone, but with a high degree of variance. Regardless, should selenium supplementation dramatically reduce either GH production or release, we considered it possible that we might detect an overall reduction in its average plasma levels as compared with those of controls. Although the values for GH levels (as well as their distributions) were as expected (*Figure 5A*), we did not detect a significant difference between samples from sodium selenite-supplemented animals and control-fed littermates. While this result suggested that the release of GH into the blood stream did not differ between groups, the possibility remained that sodium selenite supplementation might instead reduce the overall production of GH. To test this, we fed young adult (2-month-old) male mice either the control diet or one containing sodium selenite for 4 weeks. We then assessed the total production of GH by administering GH-releasing peptide 2 (GHRP-2) to mice by intraperitoneal injection 10 min prior to collecting blood samples, a technique similar to that used by Thorlacius-Ussing et al. to assess GH in rats. Of note, the interval selected (10 min) was informed by previous studies

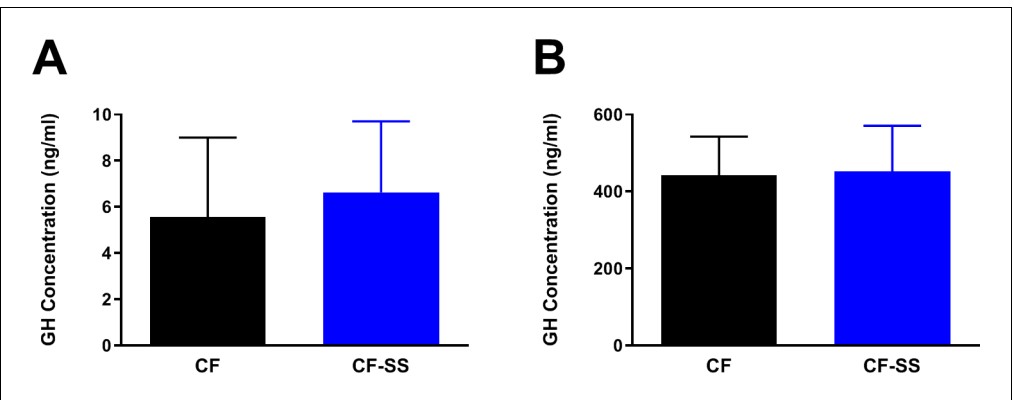

**Figure 5.** Sodium selenite supplementation is not associated with changes in plasma GH levels in male mice. Shown are comparisons of the plasma concentrations of GH in (**A**) male mice fed for 16 weeks with either the control diet (CF) or a diet containing sodium selenite (CF-SS) or (**B**) male mice fed these diets for 4 weeks and following injection with GH-releasing peptide 2 (GHRP-2) to achieve maximum circulating levels of GH. Bars denote SEM. There are no significant differences in GH levels between the two groups. For panel **A**, n = 8 for both groups. For panel **B**, n = 9 for both groups.

(*Peroni et al., 2012*) and intended to coincide with the maximum release of GH into the bloodstream. Using this approach, we found that circulating GH levels were approximately two orders of magnitude greater than those of mice not receiving GHRP-2 injections (*Figure 5A,B*), demonstrating the efficacy of this compound. Despite this, we remained unable to detect a significant difference between the levels of GH in sodium selenite-supplemented animals as compared with controls. Together, these data therefore suggest that the observed diminishment of circulating IGF-1 levels that results from sodium selenite supplementation occurs independently of effects on GH. This represents an obvious difference from the case in rats, wherein selenium supplementation produces an ~80% reduction in the levels of both GH and IGF-1 (*Thorlacius-Ussing et al., 1987*).

## Selenium supplementation using an organoselenium compound also protects mice against diet-induced obesity

Regardless of the nature of the mechanism by which IGF-1 levels are reduced by sodium selenite supplementation, it is clear that this intervention is highly effective in protecting against diet-induced obesity and conferring a multitude of additional MR-like healthspan benefits. However, for the purposes of providing dietary selenium, many studies have used organoselenium compounds rather than inorganic selenium-containing salts. For example, a large-scale clinical trial was performed to assess whether administration of selenomethionine to humans might protect against the development of prostate cancer (*Klein et al., 2011*; *Lippman et al., 2009*). To determine whether other sources of selenium might confer the benefits mentioned above, or whether this is a property unique to sodium selenite, we fed young adult (2-month-old) male mice an otherwise normal, methionine-replete high-fat diet containing 0.0037% selenomethionine (CF-SM) and assessed their body mass and food consumption over time (*Figure 6A–C*), as well as their body condition (*Figure 6D–F*) and adiposity (*Figure 6G–I*) at the end of the experiment. In addition, another group of males were fed a second diet containing a higher dose of selenomethionine (0.0073%; CF-SM 2×) and were similarly assessed. Overall, we found that the response to the selenomethionine-containing diets was similar to that of sodium selenite-containing diets, albeit somewhat less robust. For example, selenomethionine-supplemented male mice showed a dose-dependent protection against diet-induced obesity (*Figure 6A,D,G,H*), with less overall body mass than control-fed littermates (*Figure 6A,D*). As was the case for sodium selenite-supplemented animals, this difference was primarily due to lower adiposity, as we observed less inguinal and perigonadal adipose tissue in selenomethionine-supplemented male mice (*Figure 6G,H*). However, this effect did not extend to liver, as liver mass was not significantly different between experimental and control animals (*Figure 6I*). This intervention also differed somewhat from sodium selenite supplementation in that neither lean body mass nor overall

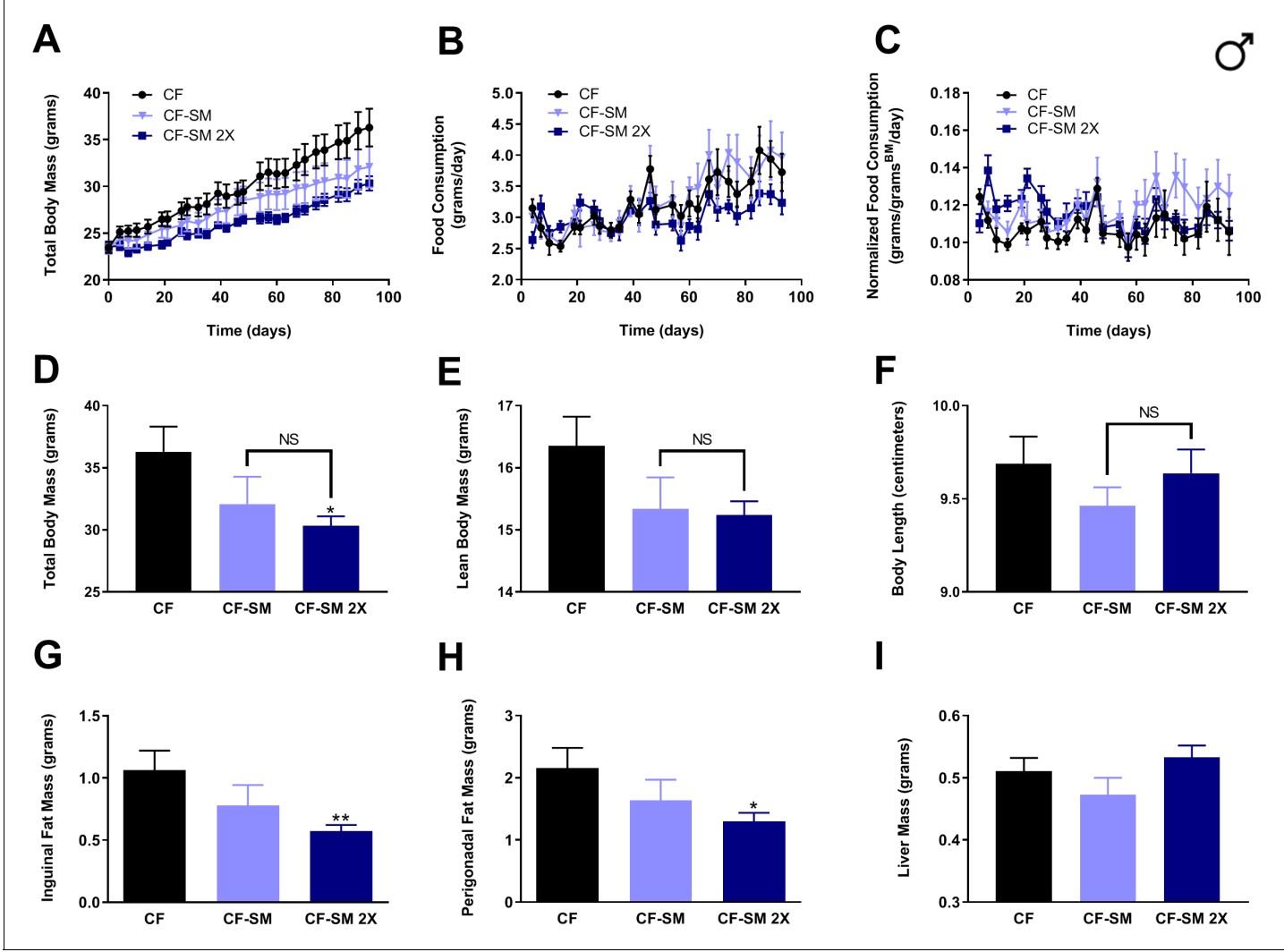

**Figure 6.** Selenium supplementation using an organoselenium compound confers a dose-dependent partial protection against diet-induced obesity to male mice. Comparisons over time of average values for (A) total body mass, (B) food consumption, and (C) food consumption normalized to total body mass for control-fed (CF; black circles) and moderately selenomethionine-supplemented (CF-SM; light blue triangles) male mice, as well as mice fed a slightly higher amount of selenomethionine (CF-SM 2×; dark blue squares). Average values at conclusion of the experiment (13.3 weeks) are also shown for (D) total body mass, (E) lean body mass, (F) body length, (G) mass of inguinal fat pads, (H) mass of perigonadal fat pads, and (I) liver mass for all feeding conditions. For all panels, bars denote SEM. For panels D–I, statistically significant differences (as compared with the corresponding CF values) are indicated (*p<0.05; **p<0.01). N = 8 for all groups.

length was found to be significantly different from control-fed male littermates (*Figure 6E,F*). As before, we confirmed that the moderate protection against diet-induced obesity observed was not due to CR, as animals ate the selenomethionine-containing diets comparably to the control diet (*Figure 6B*), and their food consumption was actually somewhat higher than that of control-fed animals, when normalized for body mass (*Figure 6C*).

To test the efficacy of selenomethionine supplementation in protecting against diet-induced obesity in females, adult (9-month-old) animals were fed either the control diet or food containing the higher dose of selenomethionine (CF-SM 2×). As before, measurements of body mass and food consumption were performed over the duration of the experiment (*Figure 7A–C*), determination of body condition was performed at termination (*Figure 7D–F*), and adiposity was assessed by weighing surgically resected fat pads and liver (*Figure 7G–I*). Overall, the results were consistent with those for males, providing further evidence that selenomethionine supplementation confers a moderate protection against diet-induced obesity. Longitudinal measurements of body mass revealed

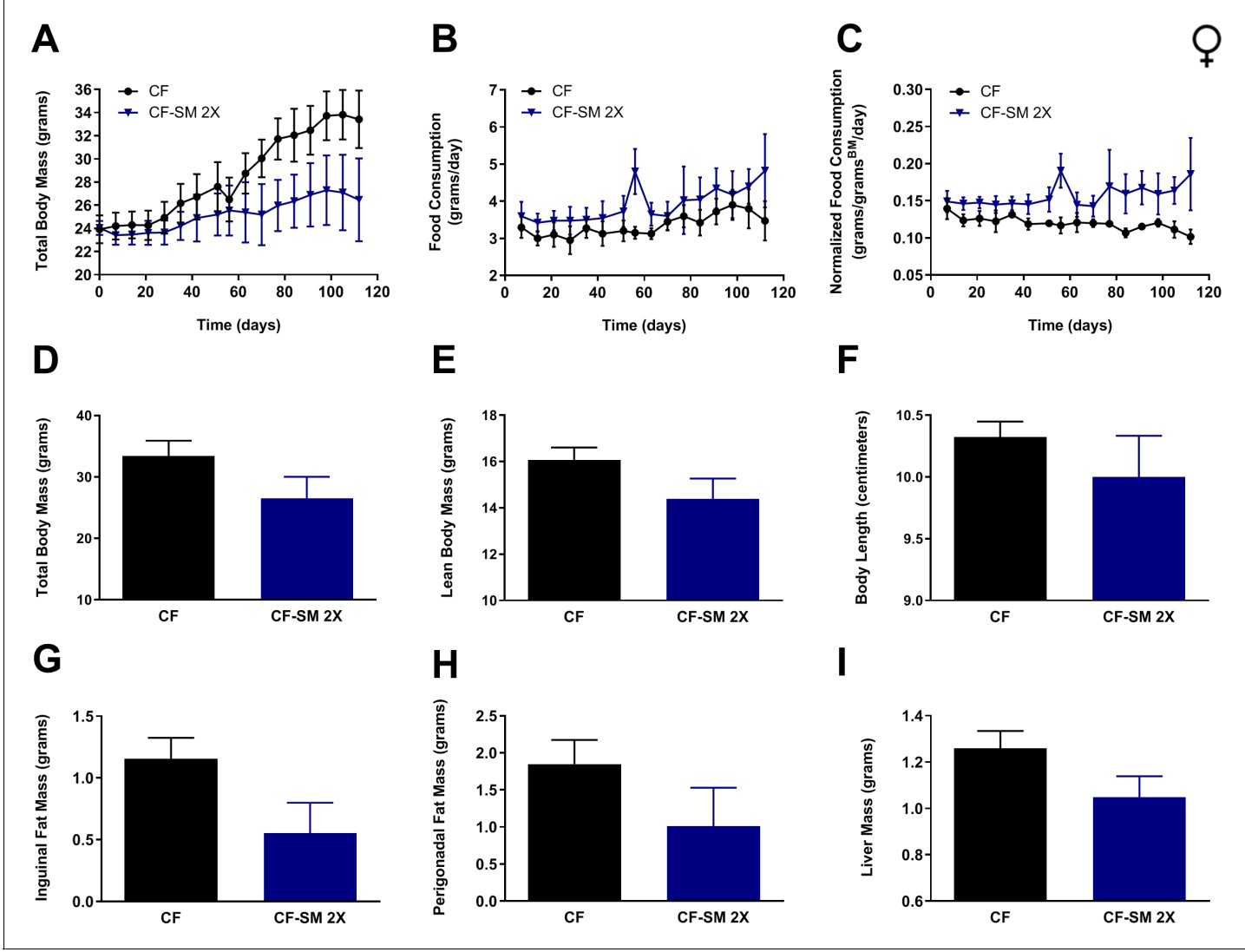

**Figure 7.** Selenium supplementation using an organoselenium compound confers partial protection against diet-induced obesity to female mice. Comparisons over time of average values for (A) total body mass, (B) food consumption, and (C) food consumption normalized to total body mass for control-fed (CF; black circles) female mice, as well as mice fed the higher of two amounts of selenomethionine used for these studies (CF-SM 2×; dark blue triangles). Average values at conclusion of the experiment (16 weeks) are also shown for (D) total body mass, (E) lean body mass, (F) body length, (G) mass of inguinal fat pads, (H) mass of perigonadal fat pads, and (I) liver mass for all feeding conditions. For all panels, bars denote standard error of the mean (SEM). N = 4 for all groups.

that selenomethionine-supplemented females gained relatively little body mass over the course of the experiment, in contrast to control-fed littermates (*Figure 7A*). However, the smaller average body size of selenomethionine-supplemented females was not found to be significantly different from control-fed animals at the terminal time-point (16 weeks; *Figure 7D*). This is likely due to a combination of animals responding less well to selenomethionine than sodium selenite and the fact that relatively few animals were available for use in this experiment (n = 4 per group). Similarly, owing to the small sample size, it cannot be said with certainty whether the lower values observed for lean body mass/length (*Figure 7E,F*), adipose tissue accumulation (*Figure 7G,H*), or liver size (*Figure 7I*) were reproducible consequences of this intervention. What is undeniable is that any benefit obtained from this intervention was not due to CR, as females found the diet to be at least as palatable as the control diet (*Figure 7B,C*). In any case, when taken together with the results for males, our results clearly demonstrate that selenomethionine supplementation can provide a degree

of protection against diet-induced obesity, suggesting that MR-like healthspan benefits may be conferred by selenium sources other than sodium selenite.

## The sodium selenite-containing diet is not toxic to mice

As the relative toxicity of selenium-containing compounds has been a subject of continual debate, there exists a significant body of work aimed at assessing the effects of selenium overnutrition (*Barceloux and Barceloux, 1999*; *Schrauzer, 2000*). The median lethal dose ($LD_{50}$) of sodium selenite for mice is ~3.4 mg/kg of body weight, nearly half that of selenomethionine (~8.8 mg/kg), and the $LD_{50}$ values for these compounds are similar for rats (*Ammar and Couri, 1981*; *Klug et al., 1952*; *Shibata et al., 1992*). However, these values were determined by intravenous injection of these compounds, rather than oral ingestion. A more recent study was performed that found the $LD_{50}$ of orally ingested sodium selenite to be 21.2 mg/kg for mice (*Wang et al., 2017*). To formulate the selenium-containing diets used for our study, we performed titrations (0.00015–0.015%; not shown) to determine the highest concentrations of sodium selenite and selenomethionine that were palatable to mice, while also remaining well below the $LD_{50}$ values for these compounds (after adjusting for food consumption and body weight). The concentrations selected were predicted to result in body weight-normalized selenium intakes roughly equivalent to that previously shown to decrease circulating IGF-1 levels in rats (*Thorlacius-Ussing et al., 1987*). At this level of supplementation, we have observed no ill effects of selenium in any of our experiments. In fact, at the time of writing, a cohort of animals has been fed the sodium selenite-containing diet continuously for 35 weeks and have remained healthy and without impairment. It was initially apparent to us that our dietary formulations were non-toxic when we observed that selenium-supplemented animals showed less adiposity than control-fed littermates (*Figure 1G,H*, *Figure 2G,H*), but similar lean body mass (or greater, in the case of males; *Figure 1E*, *Figure 2E*) compared with methionine-restricted controls. Furthermore, despite suggestions that selenium supplementation might increase the risk of insulin resistance and diabetes, we found that selenium-supplemented animals actually had improved glycemic control (*Figure 3E,F*, *Figure 4E,F*). Finally, repeated visual assessments of the overall physical condition of selenium-supplemented animals (i.e., low adiposity, good coat appearance, high activity levels and muscle mass, etc.) have also indicated a lack of toxicity, as these animals have consistently appeared healthier than not only control-fed littermates, but also methionine-restricted controls.

Nevertheless, we performed additional experiments to directly determine whether long-term feeding of the sodium selenite-containing diet might confer toxicity to mice. Toward this end, we performed a panel of liver function tests that are not only standardly used to assess generalized toxicity, but that have previously been used to assess selenium toxicity in rodents (*Raines and Sunde, 2011*). Specifically, we determined the relative plasma levels of three liver enzymes that, when elevated, are indicative of hepatotoxicity. These are: alanine transaminase (ALT), aspartate transaminase (AST), and alkaline phosphatase (ALP). We found that mice fed the sodium selenite-containing diet for 16 weeks featured plasma levels of ALT, AST, and ALP that were either lower than (ALT) or similar to (AST and ALP) those of control-fed littermates (*Figure 8A–C*). Thus, it is highly unlikely that these animals experienced any systemic toxicity. Quite the contrary, the observed reduction of circulating ALT may represent evidence that selenium supplementation ameliorates liver dysfunction caused by the high-fat diet.

Regarding a fourth different test of putative toxicity caused by selenium supplementation, dietary selenium is known to be important for the activity of the selenoprotein iodothyronine deiodinase, an enzyme that converts thyroxine (T4) into triiodothyronine (T3) (*Bianco et al., 2002*). Consequently, it has been suggested that selenium overnutrition may cause a form of subchronic toxicity marked by excessive T3 production and hyperthyroidism. We therefore measured the levels of free T3 in the plasma of mice fed either the sodium selenite-containing diet or the control diet for 25 weeks. We found no significant difference in T3 levels (*Figure 8D*). Taken together, the above observations indicate that the sodium selenite-containing diet used in the current study is not toxic to mice. Superficially, this might seem an unexpected finding, particularly in light of the work of Wang et al., who previously reported that supplementation with various selenium-containing compounds is causative of subchronic toxicity in mice (*Wang et al., 2017*). However, there are multiple possible explanations for the apparent discrepancy between these studies. First, rather than incorporating selenium-containing compounds into pelleted diets, as we have done, Wang et al. used daily oral gavage for

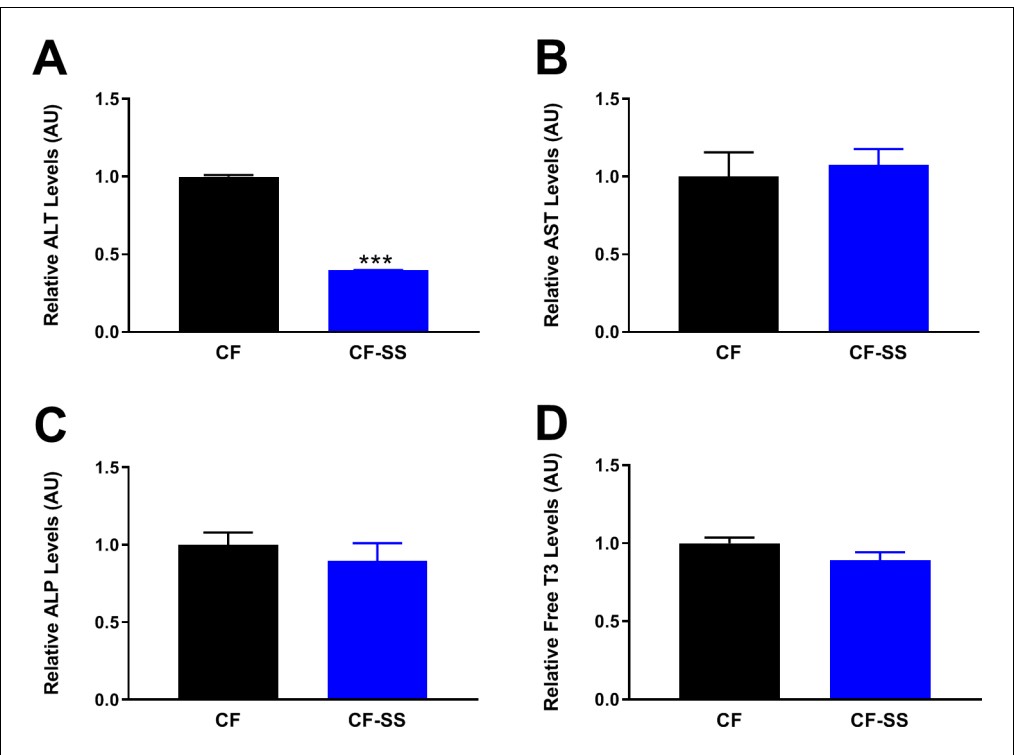

**Figure 8.** Long-term consumption of the sodium selenite-containing diet is not toxic to mice. Shown are the results of a toxicology panel comprising the relative plasma levels of liver enzymes (**A**) ALT, (**B**) AST, and (**C**) ALP, as well as the relative plasma levels of free triiodothyronine (T3; **D**). For the liver enzymes, values are shown for male mice fed for 16 weeks with either the control diet (CF) or a diet containing sodium selenite (CF-SS). Free T3 levels were assessed in the same animals fed the indicated diets for 25 weeks. Bars denote SEM. Statistically significant differences (as compared with the corresponding CF values) are indicated (***$p<0.001$). For selenium-supplemented animals, the plasma levels of ALT, AST, ALP, and free T3 are either the same as those of control-fed littermates (AST, ALP, and free T3), or lower (ALT), indicating that the sodium selenite-containing diet is not toxic to mice. N $\geq$ 8 for all groups.

administration of these compounds to mice. As a result, it is possible that the doses of sodium selenite ingested by mice in their experiments were effectively higher than similar doses in ours, owing to the sodium selenite having undergone oven-drying in our study. Second, the authors' conclusion that long-term ingestion of sodium selenite is toxic was based primarily on the assumption that impairment of weight gain is indicative of toxicity. Indeed, in both studies, sodium selenite-supplemented animals were found to maintain a constant body mass over the course of the experiments (i. e., neither increasing nor decreasing), whereas the mass of control animals increased dramatically (*Figure 1A*, *Figure 2A*; *Wang et al., 2017*). However, it is critical to note that it is not an impairment of weight gain, per se, but rather a progressive loss of body mass that is usually (and more accurately) taken as evidence of subchronic toxicity. Even then, there are several well-characterized dietary and pharmacological interventions that reduce the body mass of obese animals, but are not considered to be toxic (e.g., MR, CR, intermittent fasting, ketogenic diet, rapamycin treatment, acarbose treatment, etc.) (*Chang et al., 2009*; *Cooke et al., 2020*; *Goodrick et al., 1990*; *Harrison et al., 2019*; *Kennedy et al., 2007*; *Lees et al., 2017*; *Wang et al., 2018*; *Yu et al., 2018*). Presumably, the reduction of IGF-1 signaling that occurs in response to all of these interventions represents the shared mechanism by which they both regulate body size and confer healthspan benefits to animals. Overall, our data are consistent with selenium supplementation operating similarly, by protecting mice against diet-induced obesity through reduced IGF-1 signaling, but with no evidence of toxicity.

## Selenium supplementation extends the lifespan of yeast and requires both mitophagy and Alt1 transaminase activity

To develop a simple, genetically tractable model with which to explore the mechanistic basis of the benefits of selenium supplementation, we decided to make use of budding yeast. There are a number of assays that explore different aspects of cellular aging in yeast, including the chronological lifespan (CLS) assay, which assesses the period of time that non-dividing yeast remain viable and able to re-enter the cell cycle, as well as the replicative lifespan (RLS) assay, which measures the number of cell divisions that a newborn yeast cell can undergo (*Fabrizio and Longo, 2003*; *Longo et al., 1996*; *Mortimer and Johnston, 1959*; *Müller et al., 1980*). The former assay is intended to serve as a model of aging in quiescent eukaryotic cells, whereas the latter is used to assess how aging affects the proliferative capacity of mitotic cells. In previous work, we demonstrated that multiple interventions that produce the methionine-restricted state in yeast also dramatically extend CLS (*Johnson and Johnson, 2014*; *Plummer and Johnson, 2019*). A recent study also found that glucose restriction, which is known to extend RLS, does so by reducing the intracellular concentration of methionine (*Zou et al., 2020*). This finding is intriguing, particularly in light of unpublished studies from our laboratory (JKT; not shown) revealing that replicatively aged yeast show alterations in methionine metabolism. To determine whether selenium supplementation might, like MR, extend yeast lifespan, we assessed both the chronological and replicative lifespans of selenomethionine-supplemented yeast (*Figure 9A,B*). We found that aging wild-type yeast in selenomethionine-containing medium resulted in a 62% extension of maximum CLS (21 days vs 13 days; p=0.0164) as compared with yeast aged in control medium (*Figure 9A*). In addition, selenium supplementation also

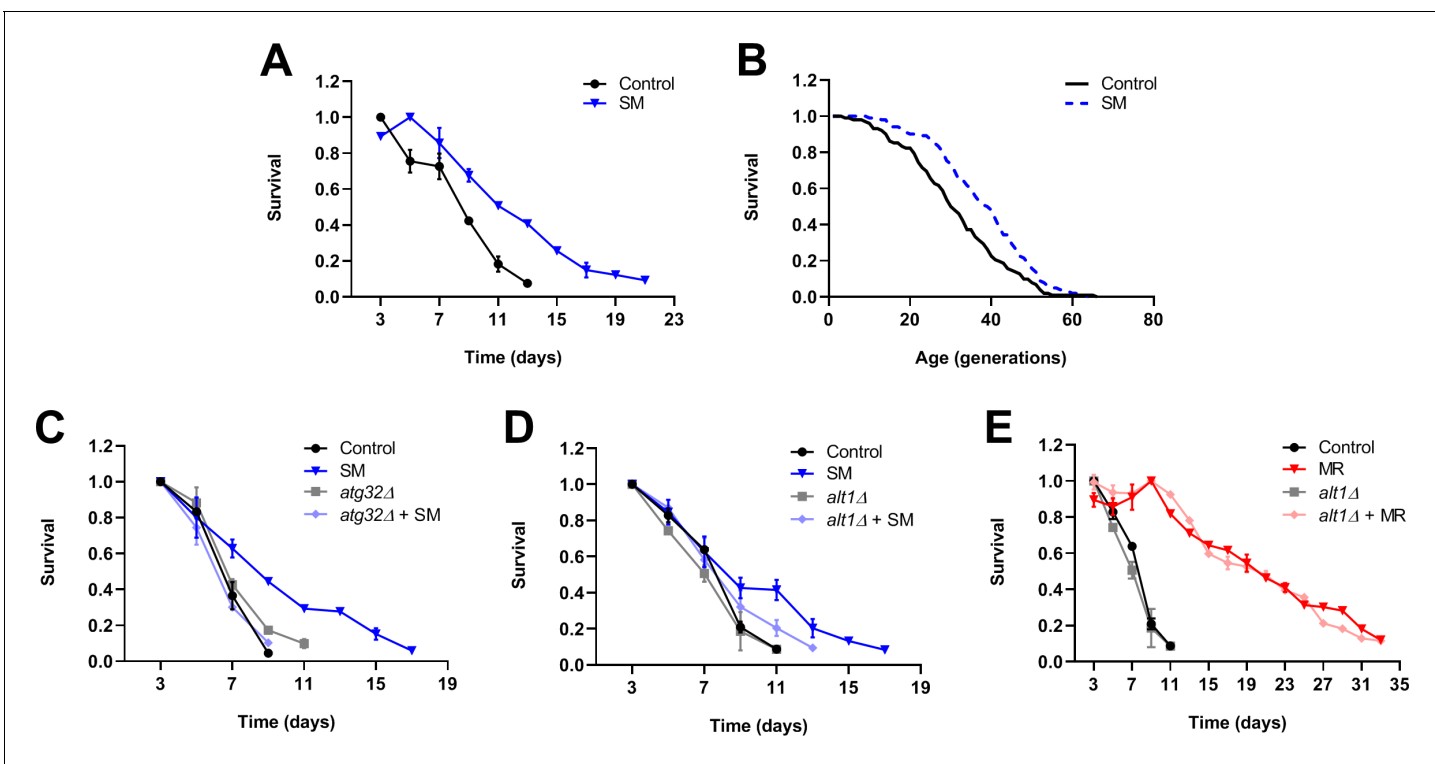

**Figure 9.** Selenium supplementation extends yeast lifespan, with chronological lifespan extension being dependent on mitophagy and Alt1 transaminase activity. Survival curves are shown for experiments assessing the effects of selenium supplementation on yeast chronological lifespan (**A**, **C–E**) and replicative lifespan (**B**). The presence of selenomethionine (SM) extends the chronological lifespan (**A**) and replicative lifespan (**B**) of wild-type yeast as compared with controls. Both the core mitophagy factor Atg32 (**C**) and the Alt1 transaminase (**D**) are required for the full extension of yeast chronological lifespan by selenium supplementation. In contrast, the Alt1 transaminase is dispensable for the extension of chronological lifespan by methionine restriction (**E**). Bars denote SEM.

The online version of this article includes the following figure supplement(s) for figure 9:

**Figure supplement 1.** Selenium supplementation decreases yeast histone deacetylase activity.

produced a dramatic extension of RLS, with median survival extended by nine generations (30%; p=0.00008) as compared with controls (*Figure 9B*). Together, these findings demonstrate that supplementing yeast medium with an organoselenium compound produces healthspan benefits detectable by not just one, but multiple assays of cellular aging.

To gain insight into the mechanisms underlying the benefits of selenium supplementation to yeast lifespan, we explored the genetic determinants of selenomethionine-dependent CLS extension. Because we previously found the autophagic recycling of mitochondria (i.e., mitophagy) to be indispensable for the extension of CLS by MR (*Plummer and Johnson, 2019*), we considered the possibility that selenomethionine-dependent lifespan extension might also require this activity. To test this, we aged both wild-type yeast, as well as cells deleted for a gene encoding an essential mitophagy factor (Atg32), in both normal and selenomethionine-containing media. Atg32-deficient cells aged in selenomethionine-containing medium failed to demonstrate the extended longevity associated with this intervention (*Figure 9C*), instead producing a lifespan curve nearly identical to that of control cells. Moreover, this short lifespan was not due to any putative non-specific sickness associated with the loss of mitophagy, as Atg32-deficient cells aged in normal medium were not shorter-lived than control cells. Together, these finding indicate that, similar to the case for MR, mitophagy is required for the extension of CLS by selenium supplementation. We were also prompted by a 2009 study by Lee et al. to explore whether transaminase activity might be required for the extension of CLS by selenium supplementation. In the aforementioned study, the researchers found that certain organoselenium compounds (including selenomethionine) can be converted by transaminases to their corresponding α-keto acids (*Lee et al., 2009*). The authors then convincingly demonstrated that these compounds were potent histone deacetylase (HDAC) inhibitors, capable of promoting histone H3 acetylation status. In turn, the observed increase in H3ac likely altered gene expression in the cultured cells used for the study. To test the hypothesis that the benefits of selenomethionine to yeast might require the conversion of this compound to its α-keto acid (α-keto-γ-methylselenobutyrate; KMSB), we assessed the CLS of both wild-type yeast and cells deleted for the gene encoding the Alt1 transaminase, aged in both normal and selenomethionine-containing media. The results revealed that transaminase activity was required for the full extension of CLS by selenium supplementation (*Figure 9D*), as Alt1-deficient cells aged in selenomethionine-containing media demonstrated only a modest extension of CLS as compared with control cells (13 days vs 11 days; p=0.014). In addition, as above, the observed impairment of selenium supplementation-dependent CLS extension was not due to any potential non-specific sickness associated with loss of Alt1 transaminase activity, as Alt1-deficient cells aged in normal medium showed a lifespan identical to that of control cells. Finally, to determine whether Alt1 transaminase activity might be needed specifically for CLS extension by selenium supplementation, or if this activity might instead be required for the extended longevity of yeast in all settings, we tested whether Alt1 deficiency compromised the extended CLS of methionine-restricted cells. We found that the CLS of Alt1-deficient cells was no different from that of wild-type cells (*Figure 9E*), indicating that Alt1 transaminase activity is not required for CLS extension, per se, but specifically for the extension of CLS by selenium supplementation.

## Discussion

One of the goals of the aging field is the identification of simple interventions that promote human healthspan. While multiple dietary interventions have been identified, including MR, that promote mammalian healthspan, a pharmacologic intervention would arguably be preferable. In the current study, we present evidence that the inorganic selenium source sodium selenite and the organoselenium compound selenomethionine confer health benefits to mice. Our data support the hypothesis that, whether engendered by selenium supplementation, MR, the GH deficiency of dwarf mice, or a multitude of other healthspan-extending interventions, such benefits are a result of decreased IGF-1 signaling. As for how impairment of IGF-1 signaling promotes healthspan, it is known that binding of IGF-1 to the IGF-1R receptor activates the PI3K/AKT/mTOR pathway, thereby stimulating growth and proliferation, while also impairing autophagy and the response to stress (*Bitto et al., 2010*; *Holzenberger et al., 2003*). It has been well established in a variety of organisms, from yeast to mammals, that both autophagy and stress response pathways are involved in the regulation of longevity (*Guarente et al., 2008*; *Postnikoff et al., 2017*; *Rubinsztein et al., 2011*; *Tyler and*

*Johnson, 2018*). Indeed, we previously found that integrated stress response factors mediate yeast RLS (*Hu et al., 2018*) and that both the retrograde stress response and the selective autophagic process mitophagy are required for the full extension of yeast CLS by MR (*Johnson and Johnson, 2014*; *Plummer and Johnson, 2019*). In addition, in the current study, we show evidence that mitophagy is indispensable for CLS extension by selenium supplementation. In mammals, not only has autophagy been implicated in the benefits of healthspan-extending interventions, but the activation of autophagy is actually sufficient to extend mouse lifespan (*Pyo et al., 2013*). Thus, reduced circulating IGF-1 likely promotes healthspan by de-repressing pro-longevity pathways downstream of PI3K/AKT/mTOR.

As for how selenium supplementation restricts the circulating levels of IGF-1 in mice, our data suggest that this may occur independently of any effects on GH. Using yeast to model the molecular mechanisms underlying selenium supplementation, we have obtained results suggesting that the transamination of organoselenium compounds (such as selenomethionine or methylselenocysteine) into their cognate α-keto acids might be required for the benefits of these molecules to yeast. For example, like the structurally similar compounds sodium butyrate and β-hydroxybutyrate (βHB), seleno-α-keto acids are potent HDAC inhibitors (*Lee et al., 2009*) and our studies demonstrate that, dependent on the presence of the Alt1 transaminase, selenium-supplemented yeast show reduced HDAC activity (*Figure 9—figure supplement 1*). An obvious possibility is that the effects of such compounds on histone acetylation, and thereby, gene expression, underlies their healthspan benefits. There might even be a precedent for such a notion, as the administration of βHB extends worm lifespan (*Edwards et al., 2014*), and both phenylbutyrate and sodium butyrate extend fly lifespan (*Kang et al., 2002*; *Zhao, 2005*). Additionally, sodium butyrate is known to ameliorate multiple age-related pathologies in mice and improves survival in a short-lived mouse model (*McIntyre et al., 2019*; *Ying et al., 2006*). Interestingly, butyrate also promotes mitochondrial biogenesis (*Uittenbogaard et al., 2018*; *Walsh et al., 2015*; *Zhang et al., 2019*). This finding raises the possibility that the seleno-α-keto acid KMSB, likely produced in selenomethionine-supplemented yeast, might also upregulate mitochondrial biogenesis, thus explaining why the extended longevity of such cells requires mitophagy. That said, beyond the effects of seleno-α-keto acids, there is an intriguing alternative mechanism by which selenium supplementation might produce the healthspan benefits that we observe. Apparently, under certain conditions, selenium can interfere with one carbon metabolism (*Speckmann and Grune, 2015*; *Speckmann et al., 2017*), which comprises the interdependent folate and methionine cycles. As a result, selenium supplementation might produce MR-like benefits by directly reducing cellular methionine levels. However, we do not favor this possibility for multiple reasons. First, in yeast studies, Alt1 transaminase activity is required for the full benefit of selenium supplementation, whereas it is entirely dispensable for MR-induced CLS extension. Second, in mouse studies, there are phenotypic differences between selenium supplementation and MR that cannot be explained by the former merely producing the latter.

Regarding such differences, as well as the similarities between selenium supplementation and MR, we find that nearly all of the MR-associated phenotypes that we tested in mice are also produced by selenium supplementation. These include (1) low levels of adiposity, (2) small overall body size, (3) protection against fatty liver, as well as altered circulating levels of (4) IGF-1, (5) FGF-21, (6) leptin, (7) adiponectin, (8) glucose, and (9) insulin. That said, a notable difference between these interventions is that only female mice alter adiponectin levels in response to selenium supplementation, whereas plasma levels of this hormone are increased by MR in both males and females. In addition, MR dramatically increases the levels of FGF-21 in all mice, regardless of their age, whereas our results indicate that FGF-21 does not change in response to selenium supplementation in young (2-month-old) animals. The latter observation is particularly fascinating, as it demonstrates that an MR-like phenotype is possible in the absence of elevated FGF-21 levels. This is consistent with a recent report that FGF-21 is actually dispensable for the healthspan benefits associated with MR (*Cooke et al., 2020*).

Importantly, both inorganic selenium sources like sodium selenite and the organoselenium compound selenomethionine have been demonstrated to be safe for human consumption and, as mentioned above, a large-scale clinical trial (selenium and vitamin E cancer prevention trial; SELECT) has been performed aimed at testing whether selenomethionine supplementation might reduce the incidence of prostate cancer in humans (*Klein et al., 2011*; *Lippman et al., 2009*). While this study failed to find an anti-tumor effect of this intervention under the conditions tested, it is possible that

such an activity might be achieved (1) by using a different dosage of selenomethionine, or (2) by using a different selenium source with greater bioavailability and/or efficacy. These possibilities are supported by the fact that, in our study, selenomethionine was less active than sodium selenite in producing MR-like benefits for mice. Another potential explanation as to why the SELECT trial failed to identify a benefit of selenomethionine supplementation is that perhaps MR-like benefits other than an anti-tumor effect were engendered by this intervention, but as such benefits were beyond the scope of the tumor-focused study, they were therefore not detected. Regardless, it has been clear for some time that selenium availability is positively correlated with human healthspan. For example, selenium levels fall with age, and studies have demonstrated that blood selenium levels are highly predictive of human longevity (*Akbaraly et al., 2005*; *Olivieri et al., 1994*; *Ray et al., 2006*). Not surprisingly, dietary selenium is essential for human health; moreover, it is required for the function of approximately two dozen selenoproteins, many of which are involved in redox homeostasis and the response to oxidative stress (*Kryukov et al., 2003*; *Rayman, 2012*). Indeed, selenium supplementation has been shown to protect cells against oxidative damage (*Yoon et al., 2002*) and selenoproteins are known to confer protection against age-associated neurodegenerative disorders, including Alzheimer's disease and Parkinson's disease (*Zhang et al., 2010*). Also, given our findings regarding mitophagy, it is interesting to note that selenium supplementation of cultured mouse neurons was shown to result in elevated levels of both mitochondrial biogenesis and markers of mitochondrial function (*Mendelev et al., 2012*).

In the current study, we present a novel mechanism by which selenium supplementation contributes to mammalian healthspan. We propose that this intervention downregulates IGF-1 signaling, thereby activating pathways that are both beneficial to healthspan and shared with MR. In the short term, this results in a total protection against diet-induced obesity. However, we expect that, in the long term, this intervention will also produce an MR-like extension of overall survival, as well as an amelioration of age-related pathologies. Should this prove to be the case, one can only hope that the pro-longevity pathways engaged by such interventions are sufficiently conserved that humans will receive similar benefits to their murine cousins.

# Materials and methods

**Key resources table**

| Reagent type (species) or resource | Designation | Source or reference | Identifiers | Additional information |
|---|---|---|---|---|
| Gene (*S. cerevisiae*) | *ATG32* | *Saccharomyces* Genome Database | SGD:S000001408 | |
| Gene (*S. cerevisiae*) | *ALT1* | *Saccharomyces* Genome Database | SGD:S000004079 | |
| Gene (*S. cerevisiae*) | *MET15* | *Saccharomyces* Genome Database | SGD:S000004294 | |
| Genetic reagent (*M. musculus*) | C57BL/6J | Jackson Laboratory | Stock No:000664 | |
| Commercial assay or kit | IGF-1 Quantikine ELISA kit | R&D Systems | Cat. No:MG100 | |
| Commercial assay or kit | Adiponectin Quantikine ELISA kit | R&D Systems | Cat. No:MRP300 | |
| Commercial assay or kit | Leptin Quantikine ELISA kit | R&D Systems | Cat. No:MOB00B | |
| Commercial assay or kit | FGF-21 ELISA kit | Millipore Corp. | Cat. No:EZRMFGF21-26K | |
| Commercial assay or kit | GH ELISA kit | Millipore Corp. | Cat. No:EZRMGH-45K | |
| Commercial assay or kit | Insulin ELISA kit | ALPCO Diagnostics | Cat. No:80-INSMS-E01 | |
| Commercial assay or kit | ALT activity assay kit | Sigma-Aldrich | Cat. No:MAK052 | |

*Continued on next page*

*Continued*

| Reagent type (species) or resource | Designation | Source or reference | Identifiers | Additional information |
|---|---|---|---|---|
| Commercial assay or kit | AST activity assay kit | Sigma-Aldrich | Cat. No:MAK055 | |
| Commercial assay or kit | Alkaline phosphatase detection kit | Sigma-Aldrich | Cat. No:APF | |
| Commercial assay or kit | Free T3 AccuBind ELISA kit | Monobind, Inc | Cat. No:1325–300 | |
| Commercial assay or kit | HDAC assay kit | Sigma-Aldrich | Cat. No:CS1010 | |
| Peptide, recombinant protein | GHRP-2 | Sigma-Aldrich | Cat. No:SML2056 | |
| Chemical compound, drug | Sodium selenite | Sigma-Aldrich | Cat. No:71950 | |
| Chemical compound, drug | Seleno-L-methionine | Sigma-Aldrich | Cat. No:S3132 | |
| Chemical compound, drug | Trichostatin A | Sigma-Aldrich | Cat. No:T1952 | |
| Chemical compound, drug | Sodium butyrate | Sigma-Aldrich | Cat. No:303410 | |
| Software, algorithm | GraphPad Prism | GraphPad Software | RRID:SCR_002798 | Version 8.0.0 |

## Animal care and feeding

All animal studies were approved by the Institutional Animal Care and Use Committee (IACUC) of the Orentreich Foundation for the Advancement of Science, Inc (Permit Number: 0511 MB). C57BL/6J mice (Stock number 000664) were purchased from the Jackson Laboratories (Bar Harbor, ME) and housed in a conventional animal facility maintained at $20 \pm 2°C$, $50 \pm 10\%$ relative humidity, with a 12 hr/12 hr light/dark photoperiod. Food and water were provided ad libitum. Upon arrival, mice were acclimatized for up to 1 week and fed Purina Lab Chow 5001 (Ralston Purina, Co.; St. Louis, MO). At the initiation of feeding studies, mice were given one of five isocaloric (5.3 kcal/g) high-fat diets, comprising 12% kcal protein, 31% kcal carbohydrate, and 57% kcal fat (Research Diets; New Brunswick, NJ). In brief, these diets were formulated as follows, (1) 0.86% methionine (control), (2) 0.12% methionine (methionine-restricted), (3) 0.86% methionine + 0.0073% sodium selenite (sodium selenite-supplemented), (4) 0.86% methionine + 0.0037% selenomethionine (selenomethionine-supplemented), and (5) 0.86% methionine + 0.0073% selenomethionine (2× selenomethionine-supplemented). Full details concerning diet compositions are given in *Supplementary file 1*. Mice were randomly assigned to each of the diet groups such that each group had a similar average body mass (i.e., weight-matched). Once assigned, no animals (or samples resulting therefrom) were removed from the study. Body mass and food consumption were monitored once a week for the duration of the study. Prior to blood collection, animals were fasted for 4 hr to establish a physiological baseline. Blood was then collected from the retro-orbital plexus, processed using EDTA-K2-coated blood collection tubes (Milian Dutscher Group; Geneva, Switzerland), and the resulting plasma was frozen and stored at −80°C until used for analysis. A portion of each blood sample was used for blood glucose determination using an Abbott Freestyle Lite glucometer and glucose strips (Abbott Diabetes Care, Inc; Alameda, CA). At the end of each study, animals were fasted and bled, as described above, and then sacrificed. Inguinal and perigonadal fat pads, as well as liver, were harvested by surgical resection, weighed, flash frozen, and stored at −80°C.

Measurements of lean body mass were performed, not including visceral mass, such that the values obtained represent the total fat-free musculoskeletal mass of the animals.

## Blood chemical tests

Enzyme-linked immunosorbent assay (ELISA) kits were obtained commercially and used to measure plasma levels of IGF-1 (R&D Systems; Minneapolis, MN), adiponectin (R&D Systems), FGF-21 (Millipore Corp.; Billerica, MA), leptin (R&D Systems), growth hormone (Millipore Corp.), and insulin

(ALPCO Diagnostics; Salem, NH). For studies exploring the putative toxicity of selenium-supplemented diets, enzyme activity assay kits were obtained commercially (Sigma-Aldrich; St. Louis, MO) and used to determine plasma levels of ALT, AST, and ALP. Free triiodothyronine (T3) levels were determined using a Free T3 AccuBind ELISA kit (Monobind, Inc; Lake Forest, CA). All tests were performed according to the manufacturers' recommendations and measured using a Molecular Devices SpectraMax M5 Microplate Reader (Molecular Devices LLC; San Jose, CA). Two technical replicates were performed for each sample and the statistical significance of the resulting values determined by unpaired two-tailed t-tests using the software package Prism 8 (GraphPad Software; La Jolla, CA).

## Total growth hormone measurements

To assess total GH production, mice were fasted for 4 hr prior to IP injection with 10 µg (200 µg/ml) of GH-releasing peptide 2 (GHRP-2) in sterile saline, which was obtained commercially (Sigma-Aldrich). Ten minutes after injection, blood samples were collected from the retro-orbital plexus and processed as described above. Using the resulting samples, plasma GH concentrations were determined by ELISA (Millipore Corp.), according to the manufacturer's recommendations, and measured with a Molecular Devices SpectraMax M5 Microplate Reader (Molecular Devices LLC). Two technical replicates were performed for each sample and statistical significance was determined by unpaired two-tailed t-tests using Prism 8 (GraphPad Software).

## Yeast strains

All experiments were performed using haploid strains derived from the BY4741/BY4742 background (*his3Δ1, leu2Δ0, ura3Δ0*) (*Brachmann et al., 1998*). Specifically, strains used were from the commercially available Yeast Knockout (YKO) Collection (GE Healthcare Dharmacon; Lafayette, CO). The YKO Collection comprises strains in which gene deletions of interest are marked by the KanMX drug resistance cassette (with the exception of *met15Δ0, his3Δ1, leu2Δ0, lys2Δ0,* and *ura3Δ0*).

## Yeast chronological lifespan assays

Chronological lifespan assays were performed as previously described (*Johnson and Johnson, 2014*), modified from the protocols of *Longo et al., 1996*. Briefly, cells were struck onto YPAD solid media from frozen stocks and allowed to grow at 30°C for 48 hr before colonies were inoculated into liquid synthetic complete (SC) medium. After an additional 48 hr of growth, aliquots were transferred into fresh SC medium at a concentration of ~$2 \times 10^5$ cells/ml and grown at 30°C. Following 3 days of growth, after the diauxic shift, aliquots were removed at 48 hr intervals and their colony forming units (CFUs) assessed on YPAD agar plates (n = 4 for each condition). For each culture, the point at which the remaining CFUs were found to be less than 10% of maximal was considered the end of lifespan. This cut-off was selected in order to avoid the potential confound of the GASP (Growth Advantage in Stationary Phase) phenotype, which is marked by the cyclical growth and death of a small population of cells (*Fabrizio and Longo, 2007*). Chronological lifespan assays were performed using SC medium formulated as follows: 0.67% yeast nitrogen base without amino acids, 2% glucose, 0.45% casamino acids, 0.01% tryptophan, 0.008% adenine sulfate, and 0.009% uridine. For experiments assessing the chronological lifespan of selenium-supplemented yeast, selenomethionine was added to SC medium to a final concentration of 0.00045%, at a ~1:20 stoichiometric ratio of selenomethionine:methionine. To assess the significance of lifespan differences between strains, 10% survival values (in days) were computed and used to perform unpaired two-tailed t-tests in Prism 8 (GraphPad Software).

## Yeast replicative lifespan assays

Replicative lifespan of virgin mother cells was determined by micromanipulation as previously described with minor modifications (*Kennedy et al., 1994*). Media used were freshly made synthetically defined complete media plates (SDC) with and without supplementation of filter sterilized selenomethionine (final concentration of 0.00061%; 1:20 selenomethionine:methionine) to cooling media before pouring the plates. Plates were prepared 2 days before the lifespan experiment and allowed to dry. BY4742 were streaked from frozen stocks for single colonies on SDC agar plates, and individual colonies were suspended in sterile distilled water, spotted and allowed to dry on SDC plates ±

selenomethionine supplementation (blinded to the selenomethionine content of the plates). Isolated cells' first daughters were used as virgin mother cells. Mother cells were maintained at 30°C during the day and 12°C at night to impede division. Data represent three replicates of 34 mother cells for each media condition. p-value was calculated using the two-sided Mann–Whitney U-test.

## Yeast histone deacetylase activity assays

To assess the effects of selenium supplementation on yeast HDAC activity, cell extracts were generated for cells grown as described above for CLS experiments (in the presence or absence of selenomethionine), with the exception that cultures were harvested at either 5 hr or 72 hr post-inoculation of pre-cultures into fresh SC medium. To serve as positive controls, cell extracts were also generated for cells grown in the presence of trichostatin A (20 μM) or sodium butyrate (5 mM). Specifically, harvested cell pellets were resuspended in 160 μl of ice-cold Yeast Lysis Buffer, formulated as follows: 25 mM Tris-HCl (pH 7.4), 0.5 mM EDTA, 137 mM NaCl, and 10 mM 2-mercaptoethanol. 100 μl of pre-chilled acid-washed glass beads were then added to each cell suspension, and the resulting mixtures were disrupted at 4°C using a Digital Disruptor Genie (Scientific Industries, Inc; Bohemia, NY). Ten cycles of disruption were performed, each cycle alternating 2 min of bead-beating at 3,000 rpm with 2 min of cooling on ice. Glass bead/lysate mixtures were then clarified by centrifugation at 4°C for 15 min at 13,000 rpm, and the resulting supernatants transferred to fresh tubes and stored at −80°C for subsequent analyses. Using these samples, measurement of HDAC activity was performed with the commercially available Histone Deacetylase Assay kit (Sigma-Aldrich). The assay was performed according to the manufacturer's recommendations and measured using a Molecular Devices SpectraMax M5 Microplate Reader (Molecular Devices LLC). Two technical replicates were performed for each sample and the statistical significance of the resulting values determined by unpaired two-tailed t-tests using the software package Prism 8 (GraphPad Software).

## Acknowledgements

We thank Arthur Cooper, Gene Ables, and Dawn Marie Riddle for helpful discussions during the preparation of this manuscript.

## Additional information

### Competing interests

Jessica K Tyler: Senior editor, *eLife*. The other authors declare that no competing interests exist.

### Funding

| Funder | Grant reference number | Author |
|---|---|---|
| Orentreich Foundation for the Advancement of Science | CCL023-CCL025 | Jay E Johnson |
| National Institutes of Health | R01 AG050660 | Jessica K Tyler |

The funders had no role in study design, data collection and interpretation, or the decision to submit the work for publication.

### Author contributions

Jason D Plummer, Formal analysis, Investigation, Methodology, Writing - review and editing; Spike DL Postnikoff, Writing - review and editing, Methodology (yeast RLS experiment), Investigation (yeast RLS experiment), Formal analysis (yeast RLS experiment); Jessica K Tyler, Supervision, Funding acquisition, Writing - review and editing; Jay E Johnson, Conceptualization, Formal analysis, Supervision, Funding acquisition, Investigation, Methodology, Writing - original draft, Writing - review and editing

### Author ORCIDs

Jessica K Tyler  http://orcid.org/0000-0001-9765-1659

Jay E Johnson  https://orcid.org/0000-0002-1267-7575

### Ethics

Animal experimentation: All animal studies were approved by the Institutional Animal Care and Use Committee (IACUC) of the Orentreich Foundation for the Advancement of Science, Inc (Permit Number: 0511MB).

### Decision letter and Author response

Decision letter https://doi.org/10.7554/eLife.62483.sa1

Author response https://doi.org/10.7554/eLife.62483.sa2

## Additional files

### Supplementary files

• Supplementary file 1. Composition of mouse high-fat diets.

• Transparent reporting form

### Data availability

All data generated or analyzed during this study are included in the manuscript and supporting files.

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
