## [Decision Letter]

**Acceptance summary:**

This article describes a careful and very thorough set of experiments studying the effects of selenium supplementation in both C57BL/6J mice and the budding yeast *S. cerevisiae*, using both selenium salts and organoselenium compounds. The authors propose that selenium dietary supplementation mimics or recapitulates many of the effects of methionine restriction (MR). Overall, these results are significant and of broad interest to the field.

**Decision letter after peer review:**

Thank you for submitting your article "Selenium Supplementation Inhibits IGF-1 Signaling and Confers Methionine Restriction-Like Healthspan Benefits to Mice" for consideration by *eLife*. Your article has been reviewed by three peer reviewers, one of whom is a member of our Board of Reviewing Editors, and the evaluation has been overseen by Matt Kaeberlein as the Senior Editor. The reviewers have opted to remain anonymous.

The reviewers have discussed the reviews with one another and the Reviewing Editor has drafted this decision to help you prepare a revised submission.

Summary:

This manuscript demonstrated that treating mice with either sodium selenite salt or the organic selenium, selenomethionine, conferred a metabolic state similar to methionine restriction. Further, selenomethionine treatment extended both replicative lifespan (RLS) and chronological lifespan (CLS) in the budding yeast. Moreover, CLS extension was dependent on mitophagy factor Atg32 and transaminase Alt1. Overall, the concept of selenium supplementation mimicking methionine restriction and providing health benefits is novel and will be of interest to *eLife* readers.

Essential revisions:

1) One reviewer raised a serious concern about subacute toxicity caused by selenium treatment in mice.

Based on the available literature, the primary concern with the results shown for inorganic selenium is that subacute toxicity may explain the reported outcomes in body weight, body composition and hormones. Directly comparing the dose and form of inorganic selenium used in this current effort to other studies that have conducted dose response studies in rodents is necessary to resolve this. In addition, a fuller reporting of the range of doses tested and the results which informed the final choice is required. Many literature reports exist in this space for comparison. For example, a safety assessment of sodium selenite shows it to slow the growth of mice and produce organ damage (doi: 10.1155/2017/3980972) and negatively affect intestinal physiology (Food Funct., 2019,10, 5398-5412). Other reports in mice show inorganic selenium to increase glucose tolerance but also increase liver fat (doi: 10.1371/journal.pone.0101315). Selenium supplementation can cause all kinds of outcomes in rodents (https://doi.org/10.1590/s2175-97902018000100139), but human studies are mostly equivocal (doi: 10.1007/s40200-019-00419-w) or fail to show benefit (e.g., the SELECT trial as mentioned in the Discussion). Because of the complexity surrounding selenium form in addition to the choice of animal model, strain and sex, the narrative claiming efficacy of inorganic selenium supplementation to promote health span requires more data than what is provided. Circulating and tissue level of Se plus biomarkers of selenium status are required (doi: 10.3390/nu7042209) and a full toxicology and pathology assessment is necessary to prove that Se supplementation in inorganic or amino acid/organic forms at the doses chosen is not toxic. Beyond this, extending this work to health span in humans is not recommended as there are clear species differences.

2) Two reviewers raised the same concern about how mechanistically selenomethionine, which is incorporated as methionine in proteins, would function as methionine restriction.

2a) Selenomethionine is transported and metabolized like methionine and is incorporated non-specifically into body proteins (not directly converted into selenoproteins). It is unclear why and how supplementation of selenomethionine would promote reduced weight since it would increase methionine supply to body tissues above the control diet, i.e., opposite in direction to MR. Once again, measurements of selenium status in serum and tissues along with circulating levels of methionine, and a toxicology evaluation of the animals are necessary.

2b) It is known that dietary selenomethionine is either metabolized to other active selenium forms or incorporated in proteins in place of methionine. For the selenomethionine supplementation experiments in mouse and yeast, to what extent is selenomethionine being incorporated into proteins? Is protein incorporation of selenomethionine required for its physiological effects?

2c) What is a plausible mechanism for selenomethionine supplementation to mimic methionine restriction?

3) Two reviewers raised the same concern about the mechanistic link between mouse and yeast experiments. One reviewer even suggests that the yeast experiments (Figure 8) be separated from this manuscript.

3a) Mechanistic study using the yeast aging model seems quite preliminary and raises many questions. Also, the mechanisms unveiled by the yeast experiment seemed unrelated and inexplicable for the effects observed in mice. Hence, the yeast experiments (Figure 8) raised many more questions than they addressed such that they might be better off to be a separated study.

3b) There appears to be some disconnect between the mouse and yeast experiments, in terms of mechanisms. Does selenium supplementation inhibit TOR signaling and causes growth inhibition? Is selenomethionine supplementation epistatic with tor1del or rapamycin treatment?

3c) The experiments in yeast should include a control group given methionine to determine if the effect of selenomethionine is due to increased Se versus increased methionine.

3d) The authors only tested selenomethionine supplementation in the yeast experiments. How about using sodium selenite? Would you see the same effects?

3e) If the model for yeast CLS extension by selenomethionine supplementation is through its effects on histone acetylation and gene expression, the levels of H3 acetylation and gene expression changes should be further examined for clues that could explain the phenotype.

3f) Ferdouse et al. showed that methionine supplementation can rescue mitochondrial defects caused by atg32del (PMID 32123515). This conclusion appears to be consistent with the observation in this manuscript. The authors should offer some discussion on this.

[Editors' note: further revisions were suggested prior to acceptance, as described below.]

Thank you for resubmitting your work entitled "Selenium Supplementation Inhibits IGF-1 Signaling and Confers Methionine Restriction-Like Healthspan Benefits to Mice" for further consideration by *eLife*. Your revised article has been reviewed by three peer reviewers, one of whom is a member of our Board of Reviewing Editors, and the evaluation has been overseen by Matt Kaeberlein.

Summary:

This article describes a careful and very thorough set of experiments studying the effects of selenium supplementation in both C57BL/6J mice and the budding yeast *S. cerevisiae*, using both selenium salts and organoselenium compounds. The authors propose that selenium dietary supplementation mimics or recapitulates many of the effects of methionine restriction (MR). Overall, these results are significant and of broad interest to the field.

The manuscript has been improved but there is remaining concern around the question of selenium toxicity in mice and the discrepancy between this report and the prior report from Wang et al. While we do not ask you to fully understand the reason for these differences, there are two specific caveats we request be explicitly addressed in the text in order to hopefully avoid confusion in the literature around selenium toxicity.

The first is the difference in strain background used here. It appears that Wang et al. used Kunming mice while you used C57BL/6. This seems like the most likely explanation and should at least be explicitly mentioned and any known differences between these strains would be useful. https://www.ncbi.nlm.nih.gov/pmc/articles/PMC5682906/

The second is the interpretation of weight loss as a beneficial response in your manuscript. It is important to note that weight loss is also a common phenotype associated with toxicity. You should explicitly mention that the weight loss observed here cannot be differentiated from weight loss seen in response to acute toxicity along with the fact that no formal toxicity studies were carried out here.

---

## [Author Response]

Essential revisions:1) One reviewer raised a serious concern about subacute toxicity caused by selenium treatment in mice.

As noted by the reviewer, previous literature has indeed suggested that the sodium selenite-supplemented diet may be toxic. However, we found this not to be the case with our particular dietary formulation. To rigorously and quantitatively assess whether the diet was toxic to mice, we performed a series of additional experiments. These included four biochemical assays, three of which are standardly performed as part of a toxicology test (i.e., determination of plasma levels of the liver enzymes ALT, AST, and ALP). These analyses found no evidence of subacute toxicity caused by the diet. Moreover, based on measurement of ALT levels, our results actually indicate that selenium-supplemented animals experience *less* hepatotoxicity than controls. The results of these studies are presented in a new Results section, entitled “The Sodium Selenite-Containing Diet is Not Toxic to Mice”, as well as in an accompanying figure, Figure 8. We have also updated the Materials and methods section with details on these experiments. Together, these additions also satisfy the suggestion of a reviewer in comment 2a that we perform “a toxicology evaluation”.

Based on the available literature, the primary concern with the results shown for inorganic selenium is that subacute toxicity may explain the reported outcomes in body weight, body composition and hormones. Directly comparing the dose and form of inorganic selenium used in this current effort to other studies that have conducted dose response studies in rodents is necessary to resolve this. In addition, a fuller reporting of the range of doses tested and the results which informed the final choice is required. Many literature reports exist in this space for comparison. For example, a safety assessment of sodium selenite shows it to slow the growth of mice and produce organ damage (doi: 10.1155/2017/3980972) and negatively affect intestinal physiology (Food Funct., 2019,10, 5398-5412). Other reports in mice show inorganic selenium to increase glucose tolerance but also increase liver fat (doi: 10.1371/journal.pone.0101315). Selenium supplementation can cause all kinds of outcomes in rodents (https://doi.org/10.1590/s2175-97902018000100139), but human studies are mostly equivocal (doi: 10.1007/s40200-019-00419-w) or fail to show benefit (e.g., the SELECT trial as mentioned in the Discussion). Because of the complexity surrounding selenium form in addition to the choice of animal model, strain and sex, the narrative claiming efficacy of inorganic selenium supplementation to promote health span requires more data than what is provided. Circulating and tissue level of Se plus biomarkers of selenium status are required (doi: 10.3390/nu7042209) and a full toxicology and pathology assessment is necessary to prove that Se supplementation in inorganic or amino acid/organic forms at the doses chosen is not toxic. Beyond this, extending this work to health span in humans is not recommended as there are clear species differences.

As suggested by the reviewer, we have now included a discussion of the LD_50_ values for sodium selenite/selenate and selenomethionine, as well as the range of doses of selenium-containing compounds that we tested in our pilot studies, the criteria for the selection of the doses used, and a comparison of the resulting intake levels as compared with a previous study.

2) Two reviewers raised the same concern about how mechanistically selenomethionine, which is incorporated as methionine in proteins, would function as methionine restriction.2a) Selenomethionine is transported and metabolized like methionine and is incorporated non-specifically into body proteins (not directly converted into selenoproteins). It is unclear why and how supplementation of selenomethionine would promote reduced weight since it would increase methionine supply to body tissues above the control diet, i.e., opposite in direction to MR. Once again, measurements of selenium status in serum and tissues along with circulating levels of methionine, and a toxicology evaluation of the animals are necessary.2b) It is known that dietary selenomethionine is either metabolized to other active selenium forms or incorporated in proteins in place of methionine. For the selenomethionine supplementation experiments in mouse and yeast, to what extent is selenomethionine being incorporated into proteins? Is protein incorporation of selenomethionine required for its physiological effects?2c) What is a plausible mechanism for selenomethionine supplementation to mimic methionine restriction?

As pointed out by a reviewer in comment 2b, incorporation into body proteins in place of methionine is just one of multiple possible fates for selenomethionine. As mentioned by the reviewer, and as we discussed in the manuscript, selenomethionine can also be converted by transamination to α-keto-γ-methylselenobutyrate (KMSB). As a result, we hypothesized that conversion of selenomethionine to KMSB might underlie the benefits conferred by this compound. We tested this possibility using the yeast chronological aging assay, and found that the full extension of yeast lifespan by selenomethionine supplementation indeed requires transaminase activity (Figure 9D). As KMSB and similar compounds can function as histone deacetylase (HDAC) inhibitors, we discussed the possibility that the mechanism by which selenomethionine might exert its MR-like beneficial effects might be the production of KMSB, the subsequent inhibition of HDAC activity, and the resulting changes in gene expression. However, prompted by reviewer concerns, we performed additional experiments aimed at testing whether HDAC activity is in fact inhibited in selenomethionine-supplemented cells. We found that, as expected, HDAC activity levels are reduced in the presence of selenomethionine. Furthermore, in yeast, this reduction is dependent on the activity of the Alt1 transaminase. The results of these studies are presented in a new figure, Figure 9—figure supplement 1, and are briefly described in the Discussion. We have also added a Materials and methods section detailing these experiments, entitled “Yeast Histone Deacetylase Activity Assays”.

3) Two reviewers raised the same concern about the mechanistic link between mouse and yeast experiments. One reviewer even suggests that the yeast experiments (Figure 8) be separated from this manuscript.3a) Mechanistic study using the yeast aging model seems quite preliminary and raises many questions. Also, the mechanisms unveiled by the yeast experiment seemed unrelated and inexplicable for the effects observed in mice. Hence, the yeast experiments (Figure 8) raised many more questions than they addressed such that they might be better off to be a separated study.3b) There appears to be some disconnect between the mouse and yeast experiments, in terms of mechanisms. Does selenium supplementation inhibit TOR signaling and causes growth inhibition? Is selenomethionine supplementation epistatic with tor1del or rapamycin treatment?

While the yeast experiments were intended to constitute facile, genetically-tractable model systems with which to explore the benefits of MR and selenium supplementation at the cellular level, we understand and completely agree with the reviewers’ statements that these studies were of limited use with respect to understanding the mechanistic basis of these interventions in mice. As a result, we have performed an additional experiment, as described above (2), in order to gain further insight into the mechanisms underlying the extension of yeast lifespan by selenomethionine (i.e., measurements of HDAC activity). In addition, we have also moved all yeast studies to the supplement, in order to make more apparent that the mouse experiments represent the main focus of the manuscript. (Note-the *eLife* editorial support staff have instructed that one of these figures be moved back into the main body of the text to conform to *eLife* style preferences for supplementary figures “We would like you to change the way in which the supplementary figures included with your submission are cited. Each figure supplement should be associated with a main figure. Please order them and cite them sequentially, e.g., Figure 1, Figure 2, Figure 2—figure supplement 1, Figure 2—figure supplement 2, Figure 3, Figure 3—figure supplement 1, and so on) and ensure any references to these figures in your response to the review comments are also updated.”

3c) The experiments in yeast should include a control group given methionine to determine if the effect of selenomethionine is due to increased Se versus increased methionine.

The amount of selenomethionine added to aging yeast cultures is vanishingly small (chronological aging, 0.00045%; replicative aging, 0.00061%) and nearly equivalent to differences in final methionine concentrations likely to be achieved simply due to stochastic variations in media preparation. Put another way, for chronological lifespan experiments, the proposed control would involve increasing the methionine concentration from 0.009% to only 0.00945%. That said, we have performed unpublished experiments featuring greater methionine supplementation (0.045%) wherein there was no effect on chronological lifespan (either positively or negatively) as compared with the standard control.

3d) The authors only tested selenomethionine supplementation in the yeast experiments. How about using sodium selenite? Would you see the same effects?

We performed pilot studies using sodium selenite, but as we found this compound to be much more bioactive than selenomethionine (similar to the results of our mouse studies), it was technically challenging to identify a “therapeutic” dose that was effective at extending yeast lifespan without also inhibiting cell growth.

3e) If the model for yeast CLS extension by selenomethionine supplementation is through its effects on histone acetylation and gene expression, the levels of H3 acetylation and gene expression changes should be further examined for clues that could explain the phenotype.

We agree with the reviewer that a further exploration of our hypothesis on the mechanistic basis of yeast lifespan extension by selenomethionine supplementation was merited. As a result, and as mentioned above, we performed an additional experiment where we measured the relative HDAC activity levels in wild-type and Alt1-deficient cells, in the presence or absence of selenomethionine (Figure 9—figure supplement 1). As expected, we found that selenomethionine supplementation reduces HDAC activity, dependent on the Alt1 transaminase. Future studies will characterize the effects of this intervention on histone post-translational modifications, as well as the resulting changes in gene expression.

3f) Ferdouse et al. showed that methionine supplementation can rescue mitochondrial defects caused by atg32del (PMID 32123515). This conclusion appears to be consistent with the observation in this manuscript. The authors should offer some discussion on this.

We agree that the findings of the aforementioned paper are consistent with the observation that the benefits of MR to yeast require Atg32. It may be that the authors of that study observed a benefit of methionine supplementation to Atg32-deficient cells because their standard culture conditions (i.e., fermentation media for sake yeast) result in low-level mitophagic activity and alterations in metabolism associated with limited methionine availability. In such a case, methionine supplementation would obviate the need for Atg32, as such cells would be less dependent on mitochondrial function. That said, such a discussion would have been more appropriate in our previous study (Plummer and Johnson, 2019 ), wherein we first demonstrated that the benefits of MR to yeast require Atg32. Unfortunately, it is beyond the scope of the current study, which explores the genetic determinants that support yeast lifespan extension by selenium supplementation only.

[Editors' note: further revisions were suggested prior to acceptance, as described below.]

[…] The manuscript has been improved but there is remaining concern around the question of selenium toxicity in mice and the discrepancy between this report and the prior report from Wang et al. While we do not ask you to fully understand the reason for these differences, there are two specific caveats we request be explicitly addressed in the text in order to hopefully avoid confusion in the literature around selenium toxicity.The first is the difference in strain background used here. It appears that Wang et al. used Kunming mice while you used C57BL/6. This seems like the most likely explanation and should at least be explicitly mentioned and any known differences between these strains would be useful. https://www.ncbi.nlm.nih.gov/pmc/articles/PMC5682906/The second is the interpretation of weight loss as a beneficial response in your manuscript. It is important to note that weight loss is also a common phenotype associated with toxicity. You should explicitly mention that the weight loss observed here cannot be differentiated from weight loss seen in response to acute toxicity along with the fact that no formal toxicity studies were carried out here.

We thank the Editors for their efforts in considering our manuscript for publication. We have addressed their final concern (an insufficient discussion of the discrepancy between a previous study {that finds that oral supplementation of selenium is toxic to mice} and the current study {that finds no evidence of toxic effects}) and have revised the manuscript accordingly. Details of the changes and our response to Editor comments are below.

1) While we appreciate the Editors’ suggestion that mouse strain differences may account for the discrepancy in the assessment of toxicity in our study as compared with that of Wang *et al.*, we do not favor this explanation as there is no evidence to suggest this may actually be the case. Instead, in the revised manuscript, we discuss how technical differences in the administration of sodium selenite may result in a greater effective dose of this compound being used in the experiments of Wang *et al.* as compared with ours. In the former study, sodium selenite was administered in aqueous solution by oral gavage, whereas in our study, this compound was included in a pelleted mouse diet that was subjected to oven-drying during production.

2) Prompted by the Editors’ comments, we have also revised the manuscript to include a thorough discussion of weight loss as a phenotype commonly associated with toxicity. Unfortunately, the Editors’ statement that “… the weight loss observed here …” is not accurate. That is, neither the current study nor the study of Wang *et al.* report weight *loss* by sodium selenite-supplemented animals. These animals actually maintain a constant weight, and thus, do not suffer the weight *gain* experienced by control animals. The confusion may have stemmed from the occasional use of the term “reduced” in the previous draft of the manuscript when comparing the body mass of selenium-supplemented animals with that of control animals. This term was not intended to convey that the body mass (or some other body characteristic) of experimental animals was reduced longitudinally, but rather, that their body mass was simply lower than that of control animals. To improve clarity and to avoid similar confusion among readers of *eLife*, we have changed all statements previously referring to “reduced” or “reductions of” body mass (when comparing cohorts of animals) to instead use terms like “smaller”, “lower”, or “less”. In addition, in the same statement, the Editors suggested that we make the statement that “… no formal toxicity studies were carried out here.” Unfortunately, this statement is also not accurate. As described in the text, assessment of circulating levels of liver enzymes ALT, AST, and ALP represents a conventional toxicity panel, and further, has been used previously to assess selenium toxicity in rodents. Nevertheless, the Editors’ statement indicated to us that these facts were poorly communicated in our previous draft. As a result, we have revised the relevant section of the text to improve clarity.